# An emission module for ICON-ART 2.0: Implementation and simulations of acetone

Michael Weimer[1,2], Jennifer Schröter[2], Johannes Eckstein[2], Konrad Deetz[2], Marco Neumaier[2], Garlich Fischbeck[2], Lu Hu[3], Dylan B. Millet[4], Daniel Rieger[2], Heike Vogel[2], Bernhard Vogel[2], Thomas Reddmann[2], Oliver Kirner[1], Roland Ruhnke[2], and Peter Braesicke[2]

[1]Steinbuch Centre for Computing, Karlsruhe Institute of Technology, 76344 Eggenstein-Leopoldshafen, Germany
[2]Institute of Meteorology and Climate Research, Karlsruhe Institute of Technology, 76344 Eggenstein-Leopoldshafen, Germany
[3]University of Montana, Department of Chemistry and Biochemistry, Missoula, MT, 59812, USA
[4]Department of Soil, Water, and Climate, University of Minnesota, Saint Paul, MN, 55108, USA

*Correspondence to:* M. Weimer (michael.weimer@kit.edu)

**Abstract.** We present a recently developed emission module for the ICON (ICOsahedral Non-hydrostatic)-ART (Aerosols and Reactive Trace gases) modelling framework. The emission module processes external flux data sets and increments the tracer volume mixing ratios in the boundary layer accordingly.

The performance of the emission module is illustrated with simulations of acetone, using a simplified chemical depletion mechanism based on a reaction with OH and photolysis only. In our model setup, we calculate a tropospheric acetone lifetime of 33 days, which is in good agreement with the literature. We compare our results with ground-based as well as with airborne IAGOS-CARIBIC measurements in the upper troposphere and lowermost stratosphere (UTLS) in terms of phase and amplitude of the annual cycle. In all our ICON-ART simulations the general seasonal variability is well represented but uncertainties remain concerning the magnitude of the acetone mixing ratio in the UTLS region.

In addition, the module for online calculations of biogenic emissions (MEGAN2.1) is implemented in ICON-ART and can replace the offline biogenic emission data sets. In a sensitivity study we show how different parametrisations of the leaf area index (LAI) change the emission fluxes calculated by MEGAN2.1 and demonstrate the importance of an adequate treatment of the LAI within MEGAN2.1.

We conclude that the emission module performs well with offline and online emission fluxes and allows the simulation of the annual cycles of emissions dominated substances.

## 1  Introduction

Many trace gases (called tracers hereafter) are emitted into the atmosphere by sources located at the Earth's surface. Especially for volatile organic compounds (VOCs), natural and anthropogenic emissions as well as secondary production from emitted precursor compounds are major atmospheric sources (e.g., Blake and Blake, 2002; Atkinson and Arey, 2003).

Different approaches to include emissions in atmospheric modelling have been developed in the past and are used in current chemistry climate models: In the limited area chemistry model WRF-chem (Grell et al., 2005) emissions are treated as pro-

duction terms in the chemical equations (McKeen et al., 1991). Emissions can be prescribed as a flux condition in the vertical diffusion, as e.g. in the Community Atmosphere Model (Lamarque et al., 2012; Neale et al., 2013) which is part of the Community Climate System Model (CCSM, Gent et al., 2011). This method is also used for emissions in the planetary boundary layer in the GEOS-Chem model (GEOS: Goddard Earth Observing System Model, Bey et al., 2001) including the HEMCO

module (Keller et al., 2014). Emissions in higher altitudes are brought into GEOS-Chem as a tendency in the respective height of the emissions. The MESSy interface (Jöckel et al., 2005) incorporated e.g. in the EMAC model (EMAC: ECHAM/MESSy Atmospheric Chemistry, Jöckel et al., 2006) gives the possibility to choose the used method for including emissions into the model: Either emissions are prescribed as flux condition as described above or the increase of the tracer mixing ratio is calculated and added to the tracer (Kerkweg et al., 2006). The latter method is also used for the MACC reanalysis (Monitoring

Atmospheric Composition and Climate, Inness et al., 2013) and in the coupled limited area model COSMO-ART (COSMO: COnsortium for SMall-scale MOdelling, ART: Aerosols and Reactive Trace Gases, Vogel et al., 2009).

Recent work also includes the development of chemistry-climate models on icosahedral grids (Suzuki et al., 2008; Elbern et al., 2010; Niwa et al., 2011; Goto et al., 2015; Rieger et al., 2015). The ICOsahedral Non-hydrostatic modelling framework (ICON) has been designed for the simultaneous usage for numerical weather prediction and climate simulations (Zängl et al.,

2015). It includes the possibility of local grid refinement (nests) with two-way interaction. Due to its good scaling properties ICON is applicable on high performance computers of the next generation.

In the previous version of the coupled chemistry climate modelling framework ICON-ART, only emissions of aerosols are considered (ART: Aerosols and Reactive Trace gases, Rieger et al., 2015). A module accounting for trace gas emissions was not existing so far.

Here we present a module for including emissions from external data sources in ICON-ART which is independent of the temporal resolution of the underlying emission data. This module reads emission mass fluxes from data sets, remapped to the unstructured ICON grid, and interpolates them to the ICON-ART simulation time. After conversion to volume mixing ratio (VMR) the emissions are added to the tracer VMR in ICON-ART in the lowest model layers. This number is specified by the user.

In addition, the Model of Emissions of Gases and Aerosols from Nature (MEGAN2.1, Guenther et al., 2012) as implemented in ICON-ART is presented. This model calculates biogenic emissions of VOCs online, i.e. dependent on the current state of the atmosphere.

We also describe a new simplified mechanism for depletion of trace gases due to reaction with OH, the main tropospheric sink for most VOCs (Blake and Blake, 2002). In addition, this mechanism includes photolysis of the species and allows the

30 space and time dependent calculation of the tracers' loss rate. Thus, these new developments now allow the investigation of VOCs with ICON-ART.

Several VOCs act as precursors of OH and $HO_2$ ($= HO_x$) radicals particularly in the dryer upper troposphere and lowermost stratosphere (UTLS) (Folkins and Chatfield, 2000). $HO_x$ can deplete ozone so that VOCs have climatic impact in the UTLS region (e.g., Neumaier et al., 2014). In this study, we will focus on the influence of acetone which is together with methanol

one of the most abundant VOC in the UTLS region. Mixing ratios of $300 - 2000\,\mathrm{pptv}$ ($1\,\mathrm{pptv} = 10^{-12}\,\mathrm{mol\,mol^{-1}}$) have been

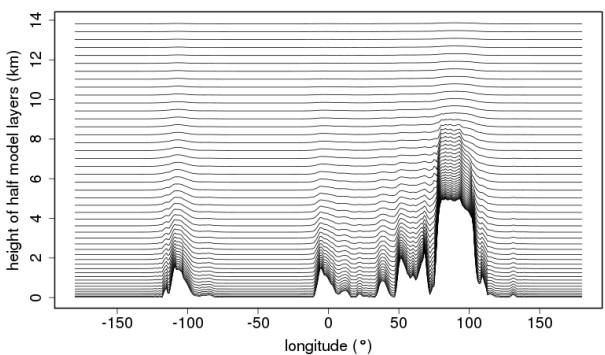

**Figure 1.** Height of the lowest 46 ICON model layers at $33°$ N in the configuration with 90 total model layers.

observed in the Northern Hemisphere mid-latitudes (Singh et al., 1995; Jaeglé et al., 1998; Heikes et al., 2002; Sprung and Zahn, 2010; Elias et al., 2011; Neumaier et al., 2014).

This study is organised as follows: In Sect. 2 the model ICON with its ART extension is described followed by the description of the emission module in Sect. 3. Then, the simplified mechanism for VOC depletion is introduced (Sect. 4). After a description
of the used measurements of acetone and the simulations for this study in Sections 5 and 6, the results are presented in Sect. 7 followed by conclusions and an outlook (Sect. 8).

## 2   The ICON model with its ART extension

In this section, we briefly describe the ICON model (Sect. 2.1) and its ART extension (Sect. 2.2). More detailed descriptions can be found in Zängl et al. (2015) and Rieger et al. (2015), respectively.

### 2.1   The ICON model

ICON is a non-hydrostatic atmospheric model developed with the aim of providing a global model for both weather and climate (Wan et al., 2013; Zängl et al., 2015). Since January 2016, it is operationally used for global numerical weather prediction at German Weather Service (DWD). In July 2016, ICON also replaced the limited area model COSMO-EU (Baldauf et al., 2011) by a nested area over Europe.

Horizontal discretisation is performed on an icosahedral-triangular C grid. In contrast to the regular latitude-longitude grid, this is an unstructured grid where the grid points are saved as one-dimensional arrays.

In this study, we use the same resolution notation as introduced by Zängl et al. (2015): R$n$B$k$ with $n$ and $k$ as indicators for root division and bisections, respectively. Usual resolutions and the corresponding global number of grid cells are shown in Table 1.

In the vertical, generalised smooth-level coordinates as described by Leuenberger et al. (2010) are used (see Fig. 1).

**Table 1.** Examples of ICON resolutions with characteristic length $\overline{\Delta x}$ and total number of cells (from Zängl et al., 2015). Characteristic length and number of cells are calculated according to $\overline{\Delta x} = \sqrt{\pi/5}\, R/(n\,2^k)$ and $n_c = 20\,n^2\,4^k$ ($R$ = Earth's radius and $n$ and $k$ as ICON resolution indicators). The grid number denotes the official ICON grid number[a] for the grid configuration used in this study, rotated by 36 degrees around z-axis.

| resolution | $\overline{\Delta x}$ (in km) | number of cells | grid number |
|---|---|---|---|
| R2B04 | 157.8 | 20 480 | 0012 |
| R2B05 | 78.9 | 81 920 | 0014 |
| R2B06 | 39.5 | 327 680 | 0016 |
| R2B07 | 19.7 | 1 310 720 | 0018 |
| R3B07[b] | 13.9 | 2 949 120 | 0022 |

[a] http://icon-downloads.zmaw.de/dwd_grids.xml
(latest access on 3 May 2017)
[b] global operational resolution at DWD

Tracers in ICON are transported by solving the continuity equation of mass for each tracer discretised with a time-split method: Finite volume method is used in the vertical whereas a simplified flux-form semi-Lagrangian method is used for horizontal transport (Miura, 2007; Lauritzen et al., 2011; Rieger et al., 2015).

Current tracers in ICON are water vapour and hydrometeors depending on the chosen microphysics scheme. In this study, the microphysics scheme is based on that used in COSMO (Doms and Schättler, 2004) and described in the technical documentation as part of the ICON source code (Seifert, 2010).

The tropopause height will play an important role in this study. In our simulations, it is calculated by ICON routines according to the thermal definition of World Meteorological Organization (WMO) (1957).

## 2.2 The ART module

The ART module for ICON is currently under development with the following aims (Rieger et al., 2015):

- Treatment of aerosols and gas-phase species in global modelling

- Gas-phase and heterogeneous chemistry

- Investigation of the feedbacks between aerosols, trace gases and the state of the atmosphere

Tracers in ICON-ART are transported and diffused in the same way as the internal ICON tracers like water vapour. The ICON-ART tracers used in this study include methane ($CH_4$), carbon monoxide ($CO$), propane ($C_3H_8$) and acetone ($CH_3C(O)CH_3$).

Chemical reactions are calculated according to the following equation:

$$\frac{\partial \overline{\rho}\hat{\psi}_i}{\partial t} = -A_i + P_i - L_i + E_i \tag{1}$$

where $\overline{\rho}$, $\psi_i$, $A_i$ and $P_i$ are Reynolds-averaged air density, partial density fraction, advection and chemical production of the tracer $i$, respectively. The hat over $\psi$ denotes the barycentric average.

$E_i$ and $L_i$ are emission and loss rate of tracer $i$, respectively. In version 1.0 of ICON-ART (Rieger et al., 2015), no general algorithm for including $E_i$ was included and the lifetime and therefore $L_i$ was assumed to be globally constant. In version 2.0 used here, we added a module for emissions (see Sect. 3) and a simplified OH chemistry for calculation of the loss rate (see Sect. 4).

Additionally, we implemented the predictor-corrector method according to Seinfeld and Pandis (2012, pp. 1125–1126) to solve Eq. (1) for tracer depletion via reaction with OH. This method is more accurate than that described by Rieger et al. (2015). A detailed description of the predictor-corrector method can be found in Appendix A.

## 3    The emission module in ICON-ART

We have included modules for offline and online calculation of emissions in ICON-ART. Both approaches are described in this section. In Section 3.1, we demonstrate our method to read and treat offline emissions whereas the description of the MEGAN2.1 model for online calculation of biogenic emissions in the configuration for ICON-ART follows in Sect. 3.2.

In order to follow the process splitting concept of ICON (Rieger et al., 2015) and to be compatible with ICON for both numerical weather prediction and climate projections the emission mass flux densities are converted to volume mixing ratio and added to the tracer volume mixing ratios.

We also perform a sensitivity study by including the emissions as lower boundary condition in the vertical turbulent diffusion scheme of ICON which can be found in the supplement of the paper. In this figure, we demonstrate that the method used in this study (see Sect. 3.1.3) and the method using the turbulent diffusion are equivalent if the emissions are included into the lowermost model layer ($n_{\mathrm{lev,emi}} = 1$). This also holds for very short lived substances such as isoprene (not shown).

### 3.1    Offline emissions

Offline emissions in ICON-ART are calculated with a new module for including emissions from external data sources which is described in the following. The process can be separated into four steps (see Fig. 2): pre-processing, initialisation, reading and finalisation.

Pre-processing (Sect. 3.1.1) is required before the model run and includes horizontal interpolation of the input data to the ICON grid as well as preparation of meta information of the data set which is committed to the module during initialisation.

The other steps are performed automatically during runtime of the model. In the step for reading emission (Sect. 3.1.2), the closest emission dates are searched and the emissions are interpolated to the current simulation time of ICON-ART. Finally, the temporally interpolated emission mass flux density is converted to VMR and added to the tracer VMR into user given number of model layers (Sect. 3.1.3).

In addition, we briefly describe the offline emission inventories used for this study (Sect. 3.1.4) and demonstrate the performance of the module (see Sect. 3.1.5).

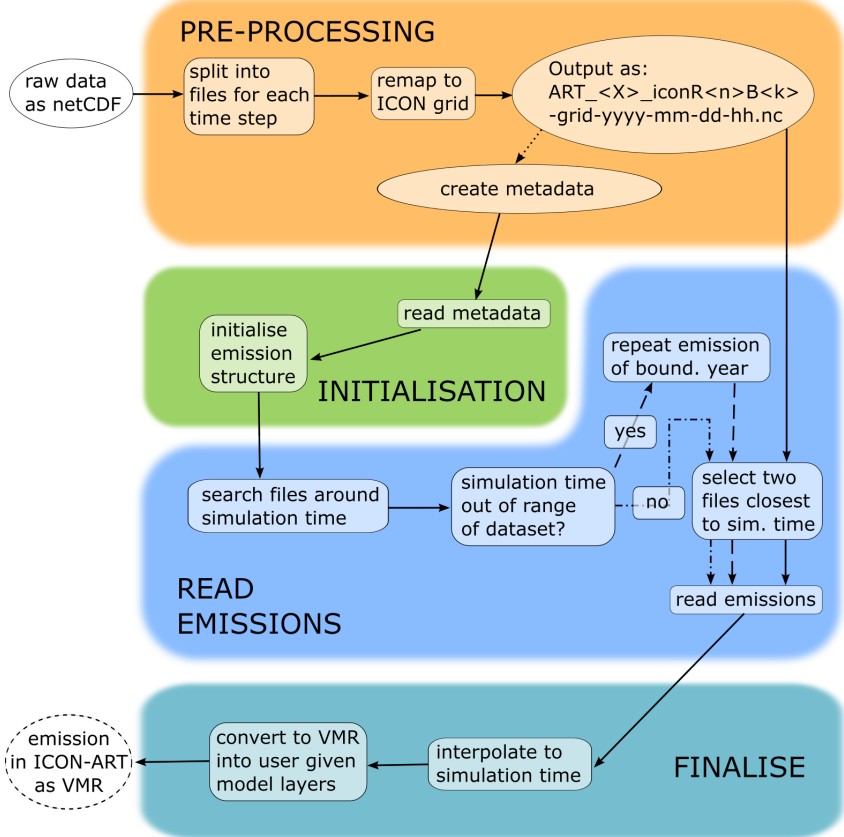

**Figure 2.** Flow chart of the process from the external netCDF emission data with regular grid and emission data as mass flux density to the emission as VMR in ICON-ART. The process can be separated into four steps: pre-processing, initialisation, read emissions and finalising the module. Pre-processing before the run of ICON-ART is necessary whereas the other processes are included in ICON-ART. Ellipses depict files while rectangles stand for processes. The different arrow lines illustrate either the interaction with the remapped netCDF data set which has to be performed by the user in the pre-processing step (dotted), the "no" path (dotted and dashed) or the "yes" path (dashed).

**Table 2.** Notation of the abbreviations used for different types of emissions denoted as X in the name structure of the files together with the corresponding integer value used in ICON-ART.

| type | abbreviation | integer value |
| --- | --- | --- |
| anthropogenic | ANT | 10 |
| biogenic | BIO | 11 |
| biomass burning | BBE | 12 |
| biogenic online | ONL | 13 |

### 3.1.1 Pre-processing of the input data and initialisation of the module

Due to the unstructured icosahedral grid of ICON (see Sect. 2.1), the usually structured latitude-longitude grid of emission data sets has to be interpolated to the ICON grid. This is managed by tools provided by DWD called the DWD ICON tools (Prill, 2016). In general, emissions are spatially highly variable. Therefore, the nearest neighbour interpolation method is applied which reasonably captures the spatial variability of the emissions. This method also conserves the total emission fluxes reasonably with a maximum deviation of $1\,\%$ in case of R2B04 and a less deviation for the other resolutions of Table 1 (not shown).

With the current version of ICON-ART, it is only possible to read files consisting of a single time step. Therefore the emission data have to be split into separate files according to their validity time.

The files to be read by the emission module have to follow the general ICON-ART name convention:

$$\text{ART\_<X>\_iconR}<n>\text{B}<k>\text{-grid-}$$
$$\text{yyyy-mm-dd-hh\_<grid-num>.nc}$$

where <X> characterises the three character abbreviation of the emission type (see Table 2), and <n> and <k> are the ICON resolution indicators in the same format as in Table 1. Additionally, the date of the emissions and the grid number (see Table 1) are part of the name structure. The maximum temporal resolution of the data set is hourly and every file can include emission data of more than one species.

Emission mass flux densities in units of $\mathrm{kg\,m^{-2}\,s^{-1}}$ are required in the raw data as the values are automatically converted to VMR after the reading process, see Sect. 3.1.3.

#### The controlling LaTeX table and "first_and_last_date.txt"

Some meta information have to be committed to the module, e.g. about the data set's location on the disk and the variable name in the remapped netCDF file for each emission data set and each tracer in ICON-ART. These metadata are controlled by a LaTeX table (see Fig. 3).

In the simplest form, each tracer in the LaTeX table is represented by one line (see tracer CO in Fig. 3). This line contains the tracer name (column 1), the number of emission types to be considered (column 2) and the standard value as mass flux

```
\begin{table}[h] \caption{Chemical Tracers and their emission folders. Type of emission: 10
    \begin{tabular}{l r c c c l}\\
    \hline
    Name      & number of types / type of emission & standard value (in kg m$^{-2}$ s$^{-1}$)
    \hline
TRCO_chemtr      & 0  &  0.00E+01  &  &  &  \\
TRCH3COCH3_chemtr & 3  &  0.00E+01  &  &  &  \\
            & 10 & 2 & 1 & acetone    & /path/to/anthropogenic/emission/data \\
            & 13 &   & 1 &            & /path/to/pft/data \\
            & 12 & 2 & 1 & acetone    & /path/to/biomass/burning//emission/data \\
```

**Figure 3.** Sample extract of a LaTeX table committing emission metadata to the module. Please note that the header lines are cut. For details of the table content see text.

density (column 3). The standard value is taken into account only if the number of emission types is zero. Then it is used as the globally applied emission mass flux density. Otherwise one line per emission type follows with empty first column, each giving the following:

- column 2: emission type as integer (see Table 2)

- column 3: number of dimensions of the emission data in the file without the time dimension: 2 or 3, for two or three dimensional data

- column 4: number of lowest model layers into which the emissions shall be included

- column 5: variable name in the netCDF files

- column 6: full path to the netCDF files

In the example of Fig. 3, no emission data sets are considered for CO. Since the standard emission value is set to zero as well, no emissions are computed for CO at all. For acetone, offline and online emissions have to be considered. The anthropogenic (type is set to 10, see Table 2) and biomass burning data set (type 12) are both two-dimensional emissions to be included in one (i.e. the lowest) model layer and with the variable name "acetone" in the netCDF files. Biogenic emissions in this example are calculated online (type 13). They are also added to the lowest model layer. The path for online emissions refers to the data set of plant functional types (see Sect. 3.2).

If the simulation time exceeds the range of the data set the boundary year is repeated as long as necessary (see Fig. 2 in the "read emission" step). That is why the boundary dates of the data set also have to be committed to the module. For this, the ASCII file "first_and_last_date.txt" placed in the same folder as the data set is used containing the first and the last date of the data set in the ICON date format in separate lines as shown in Fig. 4.

### 3.1.2 Reading emissions

The first task of the module during runtime is to find the two dates closest to the simulation time where emissions are available in the dataset. For this, one hour is successively added to or subtracted from the simulation time until a file at that date is found. The next file is searched only if the simulation time exceeds the date of the later emission file.

```
1980-01-01T00:00:00Z
2010-12-01T00:00:00Z
```

**Figure 4.** Content of "first_and_last_date.txt". It commits the boundary dates of the data set to the module with first date of data set in the first line and last date in the second one. Here, an example is given for the inventory MEGAN-MACC (see Sect. 3.1.4 for further information).

Apart from limits of the temporal resolution, no further assumptions of the data set's temporal resolution have to be made. Missing files or variable temporal resolution of the data are possible and taken care of by the model. As mentioned in Sect. 3.1.1, the lower limit of the temporal resolution is hourly. ICON-ART aborts when no file is found before or after $10^5$ hours (about 11 years) with a corresponding error message.

### 5   3.1.3   Time interpolation of the emissions and conversion to VMR

The maximum temporal resolution of the data is hourly (see Sect. 3.1.1) but the model time steps in ICON-ART are in the order of minutes for resolution R2B04 or below for higher resolutions. Therefore, the emission data is linearly interpolated to the simulation time.

After interpolation the emission mass flux density is converted to VMR. Generally, the VMR is defined as fraction of the number of moles of the tracer (in our case the number of moles of the emission $\Delta n_i$) and the number of moles of (moist) air $n_{\mathrm{air}}$:

$$\Delta X_{\mathrm{emi},i} = \frac{\Delta n_i}{n_{\mathrm{air}}} \tag{2}$$

The moles of the emission are calculated as the emission mass flux density $E_i$ multiplied by the advective model time step $\Delta t$ and the base area $A$ of the grid box and divided by the molar mass of the species $M_i$:

$$\Delta n_i = \frac{E_i A \Delta t}{M_i} \tag{3}$$

The emission flux can be included into one or more lowest model layers to be specified in the LaTeX table, see Fig. 3. In the following, we will refer to this number as $n_{\mathrm{lev,emi}}$. The total number of model layers is stated as $n_{\mathrm{lev}}$. In ICON, the lowest model layer has the highest index so that the index of the lowest model layer is $l = n_{\mathrm{lev}}$. For calculating the number of moles of the air we sum up the moles of air of the lowest $n_{\mathrm{lev,emi}}$ model layers using the ideal gas law:

$$n_{\mathrm{air}} = \sum_{l=n_{\mathrm{lev}}-n_{\mathrm{lev,emi}}+1}^{n_{\mathrm{lev}}} n_{\mathrm{air},l} = \sum_{l=n_{\mathrm{lev}}-n_{\mathrm{lev,emi}}+1}^{n_{\mathrm{lev}}} \frac{p_l V_l}{R^* T_l} = \frac{A}{R^*} \sum_{l=n_{\mathrm{lev}}-n_{\mathrm{lev,emi}}+1}^{n_{\mathrm{lev}}} \frac{p_l h_l}{T_l} \tag{4}$$

Accordingly, $p_l$, $T_l$, $h_l$ and $R^*$ stand for pressure, temperature and geometric height of the grid box and the universal gas constant, respectively.

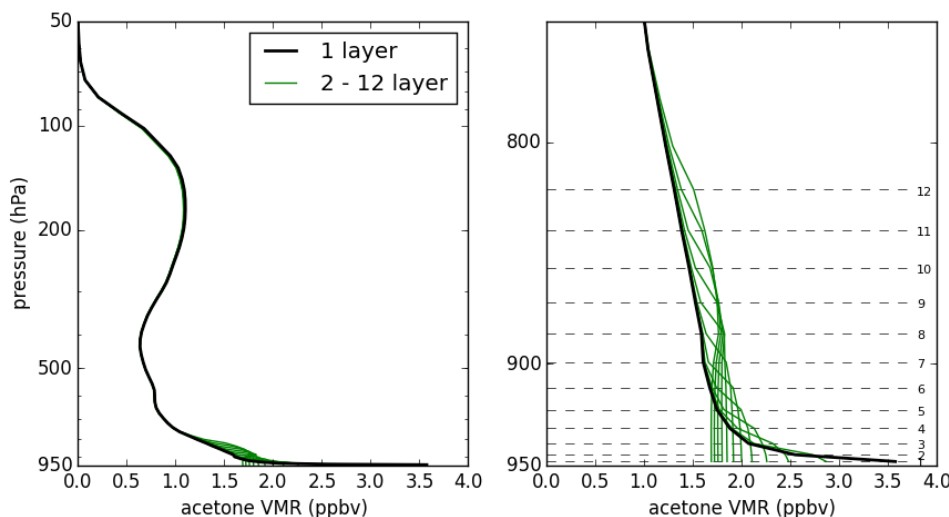

**Figure 5.** Profiles of the acetone VMR for $n_{\mathrm{lev,emi}} = 1$ (emission height of $20\,\mathrm{m}$ above ground, black thick) and 2 to 12 (emission height of $65$ to $\sim 1500\,\mathrm{m}$ above ground, green thin lines) spatially averaged over the Amazon region in Brazil on 29 February 2004, about two months after initialisation. In the right panel, the pressure range is reduced and the average height of the 12 lowest model layers are illustrated by the dashed horizontal lines.

With Eqs. (3) and (4) the VMR tendency of the emission $\mathrm{d}X_{\mathrm{emi},i}/\mathrm{d}t$, which is added to the tracer, is calculated according to:

$$\frac{\mathrm{d}X_{\mathrm{emi},i}}{\mathrm{d}t} \approx \frac{\Delta n_i}{n_{\mathrm{air}}\,\Delta t} = \frac{E_i\,R^*}{M_i}\left(\sum_{l=n_{\mathrm{lev}}-n_{\mathrm{lev,emi}}+1}^{n_{\mathrm{lev}}} \frac{p_l\,h_l}{T_l}\right)^{-1} \tag{5}$$

This method conserves mass of the emission since the calculated moles of the emission $\Delta n_i$ are independent of the choice
5 of $n_{\mathrm{lev,emi}}$ and therefore do not change if $n_{\mathrm{lev,emi}}$ is increased. The emissions are just distributed in a larger column.

To investigate the differences in changes of $n_{\mathrm{lev,emi}}$ we perform sensitivity simulations of acetone by varying $n_{\mathrm{lev,emi}}$ between 1 and 12. These simulations are based on constL(megan-off), see Sect. 6. In Figure 5, profiles of the acetone VMR are shown for the different choices of $n_{\mathrm{lev,emi}}$.

In the case of $n_{\mathrm{lev,emi}} = 1$, no emissions are included in the layers above in contrast to $n_{\mathrm{lev,emi}} > 1$. For larger values of
10 $n_{\mathrm{lev,emi}}$ the VMR in the lowermost model layer decreases subsequently since the emissions are distributed into a larger column.

Above the specified emission height, all profiles converge each other and above around $750\,\mathrm{hPa}$ the influence of varying $n_{\mathrm{lev,emi}}$ is negligible. Because of our aim to simulate acetone in the UTLS region, the choice of $n_{\mathrm{lev,emi}}$ should make no difference. That is why we simply select $n_{\mathrm{lev,emi}} = 1$ for all used offline emissions.

**Table 3.** Technical details of the emission inventories from ECCAD for tracers in ICON-ART. For abbreviations of the emission types, see Table 2.

| inventory | type | time range | resolution | | tracers | | | |
|---|---|---|---|---|---|---|---|---|
| | | | space | time | $CH_4$ | CO | $C_3H_8$ | $CH_3C(O)CH_3$ |
| MACCity[a] | ANT | 1960-2020 | $0.5°$ | month | - | ✓ | ✓ | ✓ |
| EDGARv4.2[b] | ANT | 1970-2008 | $0.5°$ | year | ✓ | - | - | - |
| MEGAN-MACC[c] | BIO | 1980-2010 | $0.5°$ | month | ✓ | ✓ | ✓ | ✓ |
| GFED3[d] | BBE | 1997-2010 | $0.5°$ | month | ✓ | ✓ | ✓ | ✓ |

[a] Lamarque et al. (2010), Diehl et al. (2012), Granier et al. (2011) and van der Werf et al. (2006),

[b] Janssens-Maenhout et al. (2011, 2013), [c] Sindelarova et al. (2014), [d] van der Werf et al. (2010)

### 3.1.4 Emission inventories

The emission data for the tracers used in this study can be downloaded from the database of Emissions of atmospheric Compounds & Compilation of Ancillary Data (ECCAD, http://eccad.sedoo.fr)[1]. The inventories used for this study are MACCity, EDGARv4.2, MEGAN-MACC and GFED3 and will be described briefly in the following paragraphs. The emission inventories are chosen according to length and temporal resolution of the data. A summary of the technical details of each used emission inventory is shown in Table 3. This table also shows which inventory is used for which tracer.

The inventory MACCity includes monthly anthropogenic emissions (Granier et al., 2011). The anthropogenic emission data are taken from the historical monthly data set of Atmospheric Chemistry and Climate Model Intercomparison Project (ACCMIP), described by Lamarque et al. (2010), and the Representative Concentration Pathways 8.5 (RCP8.5) emission scenario.

In the anthropogenic inventory Emissions Database for Global Atmospheric Research version 4.2 (EDGARv4.2, Janssens-Maenhout et al., 2011, 2013) emissions are calculated with a country-sector method based on emission factors and more than 50 categories of anthropogenic emission sources (for more information see Olivier and Janssens-Maenhout, 2015).

For the inventory MEGAN-MACC (Sindelarova et al., 2014), monthly mean biogenic emissions are calculated with MEGAN2.1 and the same 15 plant functional types as in our configuration (see Sect. 3.2). Meteorological fields are taken from the Goddard Earth Observing System (GEOS) and assimilated to model space. The leaf area index is derived from MODIS retrievals.

Biomass burning emissions in the inventory called Global Fire Emissions Database version 3 (GFED3, van der Werf et al., 2010) are calculated with a modified version of the Carnegie Ames Stanford Approach model (CASA, Potter et al., 1993; Field et al., 1995; Randerson et al., 1996). Several fire emission types are derived from satellite data and combined for calculating the carbon emission fluxes on a monthly basis in each grid cell. The emission fluxes for the substances are calculated using emission factors depending on the type of fire.

[1]latest access on 3 May 2017

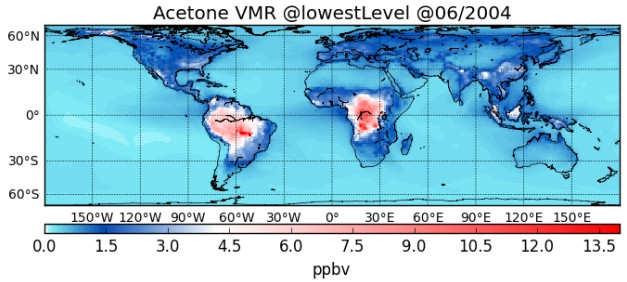

**Figure 6.** Monthly mean acetone volume mixing ratio in the lowest model layer (layer 90, height of about $20\,\mathrm{m}$ above surface) for June 2004, 6 months after initialisation of the OH-chem(megan-offl) simulation (see Sect. 6).

In the used inventories, acetone emissions are dominated by biogenic emissions. Anthropogenic and biomass burning emissions amount to $3\,\%$ and $5\,\%$ of the total global acetone emission, respectively. These values are consistent with the values published by Jacob et al. (2002) and Fischer et al. (2012).

### 3.1.5 Performance of the offline module

We demonstrate the performance of the module by including offline emissions for acetone as described in Table 3. Figure 6 shows the monthly mean acetone VMR in the lowest model layer for June 2004 in the OH-chem(megan-offl) simulation, see Sect. 6. As biogenic emissions dominate the acetone emissions, the maximum values in the acetone VMR occur over Central Africa and South America where the biogenic emissions of the inventory MEGAN-MACC also are maximised (not shown).

### 3.2 Online biogenic emissions: MEGAN2.1

To account for the influence of temperature, vegetation and photosynthetically active radiation (PAR) on the emissions of acetone, the Model of Emissions of Gases and Aerosols from Nature version 2.1 (MEGAN2.1, MEGAN-Online hereafter) (Guenther et al., 2012) is implemented into ICON-ART. In contrast to the external acetone data sets (here MEGAN-MACC) which are given as monthly mean values, the online calculation of acetone emissions within Guenther et al. (2012) allows to account for the current conditions in meteorology (especially the diurnal cycle) and vegetation. The parametrisation of biogenic emissions including acetone is described in detail in Guenther et al. (2012), therefore we present here only the main concept of the parametrisation, the changes we have made and the input provided for MEGAN-Online.

MEGAN-Online estimates the biogenic emission mass flux density $E$ in $\mathrm{\mu g\,m^{-2}\,h^{-1}}$ of the compound class $c$ via the following equation:

$$E_c = \gamma_c \sum_j \epsilon_{c,j}\, \chi_j, \tag{6}$$

where $\epsilon_{c,j}$ is the emission factor depending on the vegetation type $j$ with the fractional grid box coverage $\chi_j$. The emission activity factor $\gamma_c$ accounts for environmental and phenological conditions which affect the emissions.

MEGAN-Online includes 19 compound classes but the study on hand will focus on acetone ($c = 15$). Guenther et al. (2012) consider the emission affecting processes due to light, temperature, leaf age, soil moisture, leaf area index (LAI) and $CO_2$ inhibition. The implementation in ICON-ART only accounts for the emission responses from light, temperature, LAI and leaf age.

The light is provided by ICON-ART as photosynthetically active radiation (PAR) and temperature in the lowest model layer is a standard meteorological variable of ICON-ART. The LAI is based on external parameters read during initialisation of ICON-ART. The leaf age considers the fraction of new (FNEW), growing (FGRO), mature (FMAT) and senescing (FSEN) leaves. Due to missing information about the global distribution of these four leaf types, we assumed a uniform distribution. In addition to the standard LAI we have included the parametrisation of Dai et al. (2004) to derive $LAI_{\mathrm{sun}}$, the LAI that is lit by sun and relevant for the emissions of biogenic VOCs:

$$LAI_{\mathrm{sun}} = \frac{1}{k_b} \left(1 - \exp\left(-k_b \, LAI\right)\right) \tag{7}$$

with $k_b = G(\mu, \theta)/\mu$. The function $G(\mu, \theta)$ depends on the cosine of the solar zenith angle $\mu$ and an empirical parameter $\theta$ related to the leaf angle distribution. In the following we assume a random distribution of leaf angles which leads to $G(\mu, \theta) = 0.5$ (Dai et al., 2004). The solar zenith angle is provided by ICON-ART. $LAI_{\mathrm{sun}}$ was added to MEGAN-Online because Dai et al. (2004) have shown that the net photosynthetic rate of sun leaves is relatively high due to light saturation whereas a drastic reduction of the photosynthetic rate is visible in the low light layers of shaded leaves. With $LAI_{\mathrm{sun}}$ we therefore want to avoid an overestimation of the biogenic emissions especially in areas with high LAI which is linked to a high layering of the leaves (e.g. tropical rain forest).

To consider the vegetation type we use the external plant functional type (PFT) data set provided by CCSM (Lawrence and Chase, 2007) for 2005 with a grid mesh size of $0.05°$. This PFT data set follows the vegetation class definition of Guenther et al. (2012). The main idea of using PFTs instead of classical vegetation types is to cluster vegetation types with similar biogenic emission characteristics into the same groups for which then the emission factors $\epsilon_{c,j}$ can be derived.

In addition, MEGAN-Online needs averaged information about PAR and leaf temperature. Highest acetone emissions are observable in tropical regions and therefore we have estimated these values according to this climate zone. The mean Photosynthetic Photon Flux Density (PPFD) over $24\,\mathrm{hours}$ (PPFD24) and $240\,\mathrm{hours}$ (PPFD240) is estimated to $400\,\mathrm{\mu mol\,m^{-2}\,s^{-1}}$ from a simulation study. The mean leaf temperature over $24\,\mathrm{hours}$ (T24) and $240\,\mathrm{hours}$ (T240) is estimated to $297\,\mathrm{K}$ also based on a simulation study. The above mentioned values are not available as regular variables in ICON-ART and therefore have to be estimated (spatiotemporally constant). This could be a further source of uncertainty among the overestimation of the LAI. A sensitivity study by varying 24 and 240 h averages of PAR and leaf temperature results in changes of the emissions up to $13\,\%$ in the maximum for the ranges of 0 to $800\,\mathrm{\mu mol\,m^{-2}\,s^{-1}}$ of PAR and within the temperature range of 283 to 296 K (not shown).

**Table 4.** Parameters for MEGAN-Online used for this study. Time dependent parameters are written in italic letters.

| Variable/Parameter | Unit | Selection in ICON-ART | Meaning |
|---|---|---|---|
| $T$ | K | Standard ICON-ART output | Temperature at lowest model layer |
| $PAR$ | $\mathrm{W\,m^{-2}}$ | Standard ICON-ART output | Photosynthetically active radiation |
| $SZA$ | degrees | Standard ICON-ART output | Sun zenith angle |
| $LAI$ | $\mathrm{m^2\,m^{-2}}$ | External data from EXTPAR | Leaf area index |
| PFT | 1 | External data from CCSM | Plant functional type |
| PPFD | $\mathrm{\mu mol\,m^{-2}\,s^{-1}}$ | Derived from PAR | Photosynthetic Photon Flux Density |
| PPFDS | $\mathrm{\mu mol\,m^{-2}\,s^{-1}}$ | 125 | Standard conditions for PPFD averaged over last 24 h |
| PPFD24 | $\mathrm{\mu mol\,m^{-2}\,s^{-1}}$ | 400 | PPFD averaged over last 24 h |
| PPFD240 | $\mathrm{\mu mol\,m^{-2}\,s^{-1}}$ | 400 | PPFD averaged over last 240 h |
| T24 | K | 297 | Average leaf temperature of the past 24 h |
| T240 | K | 297 | Average leaf temperature of the past 240 h |
| FNEW | 1 | 0.25 | Fraction of new foliage |
| FGRO | 1 | 0.25 | Fraction of growing foliage |
| FMAT | 1 | 0.25 | Fraction of mature foliage |
| FSEN | 1 | 0.25 | Fraction of senescing foliage |
| G | 1 | $0.5^a$ | function for $LAI_\mathrm{sun}$ depending on $SZA$ and leaf angle distribution |

$^a$ value given by Dai et al. (2004)

For standard conditions, we use the average Photosynthetic Photon Flux Density (PPFDS) of the values given by Guenther et al. (2012): $\mathrm{PPFDS} = 125\,\mathrm{\mu mol\,m^{-2}\,s^{-1}}$. Table 4 summarises the input of MEGAN-Online and the parameter selection as used for this study.

In the following we compare the results from three emission scenarios: MEGAN-MACC, MEGAN-Online $LAI$ and MEGAN-Online $LAI_\mathrm{sun}$. MEGAN-MACC uses the emissions from the external data set. The MEGAN-Online scenarios use the online calculated emissions by using $LAI$ (MEGAN-Online $LAI$) and the LAI that is lit by sun (MEGAN-Online $LAI_\mathrm{sun}$).

Figure 7 shows the results of the three emission scenarios based on simulations using the numerical weather prediction physics package. The biogenic emission inventory MEGAN-MACC consists of monthly mean values of the MEGAN2.1 model (see Sect. 3.1.4). Therefore, the diurnal cycle is neglected in the inventory. The time series in Fig. 7 are spatially averaged over South America where the global maximum of biogenic emissions occurs, see Fig. 6. The inventory MEGAN-MACC, represented by the red dashed line in Fig. 7, is linearly interpolated between June and July. However, as acetone is emitted as by-product of photosynthesis (Jacob et al., 2002), the diurnal cycle in the emission should be considered.

With online emissions, it is now possible to capture the diurnal cycle in the emissions of acetone. The acetone online emissions are non-zero during the night which is consistent with the literature (e.g., Shao and Wildt, 2002).

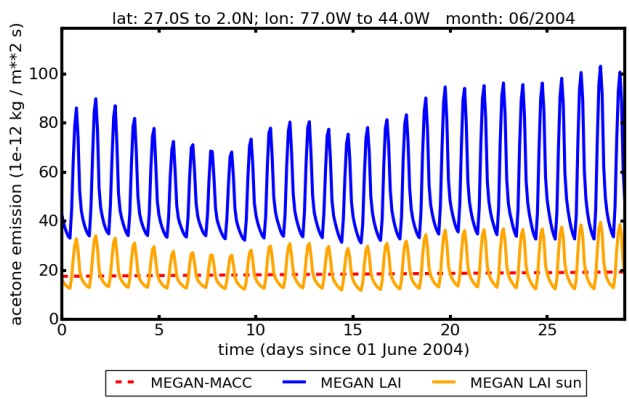

**Figure 7.** Acetone emission comparison of MEGAN-MACC (red dashed), MEGAN-Online $LAI$ (blue) and MEGAN-Online $LAI_{\mathrm{sun}}$ (orange) averaged over South America (77 to $44^\circ$ W and $27^\circ$ S to $2^\circ$ N) in June 2004.

**Table 5.** Global acetone emission flux $F$ ($\mathrm{Tg\,yr^{-1}}$) for the scenarios of Fig. 7 calculated for the year 2004. Prescribed anthropogenic and biomass burning emissions are included and account for 1.2 and $2.2\,\mathrm{Tg\,yr^{-1}}$, respectively.

| scenario | $F$ |
|---|---|
| | (in $\mathrm{Tg\,yr^{-1}}$) |
| MEGAN-MACC | 41 |
| MEGAN-Online $LAI$ | 92 |
| MEGAN-Online $LAI_{\mathrm{sun}}$ | 42 |

The emissions of the MEGAN-Online $LAI$ scenario are more than twice higher than that of MEGAN-MACC. In contrast to this, the emissions due to $LAI_{\mathrm{sun}}$ of Eq. (7) have the same global flux as MEGAN-MACC (see Table 5), considering the uncertainties in MEGAN-MACC. They are also comparable with the emission fluxes mentioned e.g. in Jacob et al. (2002), Fischer et al. (2012), Guenther et al. (2012) and Khan et al. (2015). This means that the parametrisation according to Eq. (7) can be used for investigation of the effect of the diurnal cycle on the emissions and the acetone VMR in the atmosphere for future simulations.

In order to investigate the influence of the parametrisation of $LAI$ by Eq. (7) we show in Fig. 8 the distributions of $LAI$ and $LAI_{\mathrm{sun}}$, together with its influence on the acetone emission. As expected, large values in $LAI$ (top panel) occur over the Amazon region in South America as well as in Central Africa where also the acetone VMR in Fig. 6 maximises. In addition, the forest areas in the east of Canada, northern Europe and Siberia show large values of the $LAI$. In these regions, the $LAI$ is in the order of 3 to $6\,\mathrm{m^2\,m^{-2}}$.

For the used solar zenith angle of $10.3^\circ$, the parametrisation according to Eq. (7) smoothes and reduces the LAI to values around $1\,\mathrm{m^2\,m^{-2}}$ (Fig. 8B). Only for the less vegetated regions such as deserts (Sahara or Atacama), the distribution of $LAI_{\mathrm{sun}}$ shows nearly no response to the parametrisation of Dai et al. (2004).

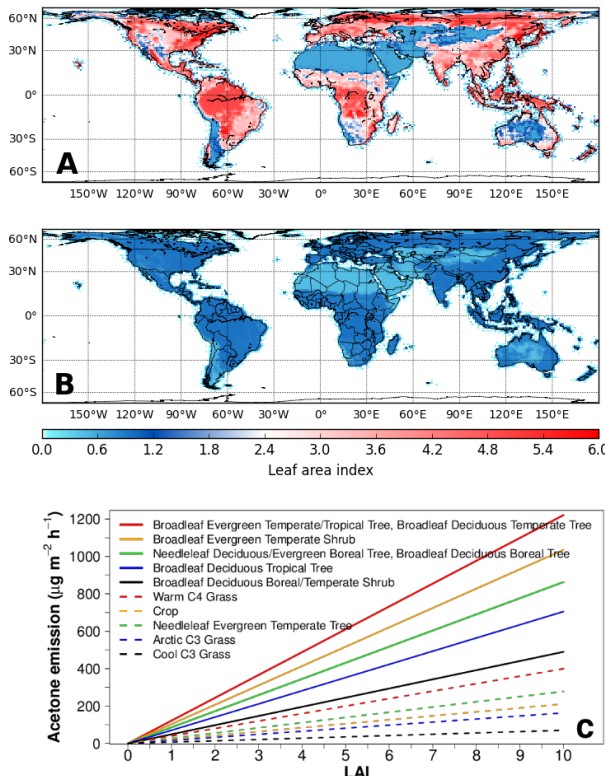

**Figure 8.** Distribution of (A) $LAI$ in ICON and (B) $LAI_{\mathrm{sun}}$ according to Eq. (7) for a solar zenith angle of $10.3°$, together with (C) dependence of the acetone emission mass flux density on the LAI for different vegetation types in MEGAN-Online with $T = 300\,\mathrm{K}$ and $PPFD = 400\,\mathrm{\mu mol\,m^{-2}\,s^{-1}}$.

In the MEGAN model the emission mass flux density is proportional to $LAI$ (Guenther et al., 2012). That is why the resulting emissions in MEGAN-Online depend linearly on the LAI for each shown plant type (Fig. 8C). The highest sensitivity on LAI can be seen for broadleaf trees in the tropics. Thus, the parametrisation of the LAI according to Dai et al. (2004) can lead to a reduction of the emissions in the order of factor 2 to 3 in these regions.

5     To conclude, the correct treatment of LAI is crucial to get realistic results of the emissions in MEGAN. The parametrisation according to Dai et al. (2004) leads to emission flux densities in the same order of magnitude as in the offline data set MEGAN-MACC (see Fig. 7). Further investigation of this will be presented in Sect. 7.

## 4   Parametrisation of tracer depletion with simplified OH chemistry

The main atmospheric sink for VOCs is the reaction with OH. Here, we illustrate the new OH depletion mechanism as
10   implemented in ICON-ART. This parametrisation calculates the loss rate of the tracers dependent on space and time and

can replace the globally constant lifetime as mentioned in Rieger et al. (2015). As an example, we illustrate the mechanism with acetone as one member of the VOCs.

## 4.1 Troposphere and UTLS region

As the tracer depletion mechanism by reaction with OH, described below, includes photolysis of ozone we first explain how
photolysis rates are treated in ICON-ART.

Photolysis rates in ICON-ART are calculated by the photolysis module which provides precise online calculation of 72 photolytic reactions including an interface between ICON, ICON-ART and the Cloud-J package (Prather, 2015). The impact of clouds and aerosols can be taken into account via different approaches implemented in the module and within Cloud-J. Cloud properties like cloud water path and effective radius of cloud droplets are calculated using ICON micro-physical properties.
Cross sections and quantum yields are given in a tabulated form originating from Sander et al. (2011) and interpolated on given pressure and temperature values of Cloud-J. The overhead ozone column, that is used, is based on the climatology of Global and regional Earth-system (Atmosphere) Monitoring using Satellite and in-situ data (GEMS, Hollingsworth et al., 2008).

The photolysis module covers roughly the wavelength region from $170\,\mathrm{nm}$ up to $850\,\mathrm{nm}$, binned into 18 wavelength bins. Thus, it is possible to accurately calculate photolysis rates from the troposphere up to the stratosphere. For the simulations
within this study the average cloud mode of Cloud-J is used.

The tropospheric OH concentration is calculated according to a simplified model, shown e.g. by Jacob (1999), see Reactions (R1) to (R8). In this model, ozone is photolysed producing an oxygen atom in excited state, $O(^1D)$. $O(^1D)$ either is quenched by collision with nitrogen ($N_2$) or oxygen ($O_2$) or reacts with $H_2O$, leading to two OH radicals:

$$O_3 + h\nu \xrightarrow{J_{O_3}} O(^1D) + O_2 \tag{R1}$$
$$N_2 + O(^1D) \xrightarrow{k_{N_2}} O(^3P) + N_2 \tag{R2}$$
$$O_2 + O(^1D) \xrightarrow{k_{O_2}} O(^3P) + O_2 \tag{R3}$$
$$H_2O + O(^1D) \xrightarrow{k_{H_2O}} 2OH \tag{R4}$$

OH is depleted by reaction with either $CH_4$ or CO, the main sinks for OH (Jacob, 1999):

$$OH + CH_4 \xrightarrow{k_{CH_4}} H_2O + CH_3 \tag{R5}$$
$$\longrightarrow \cdots \longrightarrow CO + HO_2 \tag{R6}$$
$$OH + CO \xrightarrow{M, k_{CO,1}} H + CO_2 \tag{R7}$$
$$OH + CO \xrightarrow{M, k_{CO,2}} HOCO \tag{R8}$$

Reaction rates and photolysis rates in this study are denoted as $k$ and $J$, respectively. In the following, squared brackets stand for number concentration of the species (molecules per volume unit). According to the reaction system above, the steady

state OH concentration is calculated by the following equation (cf. Jacob, 1999; Dunlea and Ravishankara, 2004; Elshorbany et al., 2016):

$$[OH] = \frac{2\,[O(^1D)]\,k_{H_2O}\,[H_2O]}{k_{CH_4}\,[CH_4] + (k_{CO,1} + k_{CO,2})\,[CO]} \tag{8}$$

where $[O(^1D)]$ is calculated by assuming a steady state with Reactions (R1) to (R4) resulting in the following formula:

$$[O(^1D)] = \frac{J_{O_3}[O_3]}{k_{O_2}\,[O_2] + k_{N_2}\,[N_2] + k_{H_2O}\,[H_2O]} \tag{9}$$

In Equations (8) and (9), the $O_3$ photolysis rate $J_{O_3}$ is calculated by the online photolysis module in ICON-ART (see above in this section). Ozone is provided by the GEMS climatology (Hollingsworth et al., 2008). $[H_2O]$ is calculated as part of the ICON micro-physics (see Sect. 2.1). $O_2$ and $N_2$ VMRs are set to $20.946\,\%$ and $78.084\,\%$, respectively (Brasseur and Solomon, 1995), and converted to number concentrations. The reaction rates in Eqs. (8) and (9) are taken from Sander et al. (2011).

With Equation (8), the loss rates of CO, $CH_4$ and $C_3H_8$ are calculated as follows:

$$L_i = k_i\,[OH], \quad i \in \{CO, CH_4, C_3H_8\} \tag{10}$$

Reaction (R5) results in a cascade of fast reactions and finally in a production of CO and is the largest source for atmospheric CO (Jacob, 1999; Boucher et al., 2001; Seinfeld and Pandis, 2012, pp. 46–47). Since Reaction (R5) is the reaction with lowest reaction rate of this cascade the chemical production of CO can be estimated as follows:

$$P_{CO} = k_{CH4}\,[OH]\,[CH_4] \tag{11}$$

As an example, we will focus on acetone in the following. Acetone is depleted either by reaction with OH or by photolysis where two channels have to be considered:

$$CH_3C(O)CH_3 + OH \xrightarrow{k_{acetone}} \text{Products} \tag{R9}$$

$$CH_3C(O)CH_3 + h\nu \xrightarrow{J_{acetone,1}} CH_3CO + CH_3 \tag{R10}$$

$$CH_3C(O)CH_3 + h\nu \xrightarrow{J_{acetone,2}} 2CH_3 + CO \tag{R11}$$

Reaction (R9) has different channels and is abbreviated here. For the reaction rate $k_{acetone}$, we use the recommended formula of Sander et al. (2011).

Following Reactions (R9) to (R11), the loss rate of acetone is determined by:

$$L_{acetone} = k_{acetone}\,[OH] + J_{acetone,1} + J_{acetone,2} \tag{12}$$

We use the mass-weighted mean shown by SPARC (2013) to calculate the lifetime of acetone:

$$\tau_{\text{acetone}} = \frac{\int [\text{CH}_3\text{C(O)CH}_3] \, dV}{\int L_{\text{acetone}} [\text{CH}_3\text{C(O)CH}_3] \, dV} \tag{13}$$

Additionally, the chemical production of acetone due to reaction of propane ($\text{C}_3\text{H}_8$) with OH is considered:

$$P_{\text{acetone}} = 0.736 \, [\text{C}_3\text{H}_8] \, [\text{OH}] \, k_{\text{C}_3\text{H}_8} \tag{14}$$

where $k_{\text{C}_3\text{H}_8}$ is the reaction rate of $\text{C}_3\text{H}_8 + \text{OH}$. The value 0.736 is a result of the two channels of this reaction and is taken from Atkinson et al. (2006).

Besides emissions, Eq. (14) is another important source for atmospheric acetone (e.g., Jacob et al., 2002). The acetone production due to other VOCs is neglected.

## 4.2 Above the UTLS region

The reaction system, described in Sect. 4.1, is valid in the troposphere, only (Jacob, 1999).

In the stratosphere, the lower VMRs of CO and $\text{CH}_4$ in Eq. (8) lead to increases of OH up to $10^8 \, \text{molec cm}^{-3}$ in the highest model layer (about $2\,\text{Pa}$). According to Brasseur and Solomon (1995), however, the OH number concentration in this altitude is in the order of $10^6 \, \text{molec cm}^{-3}$. This overestimation of the OH concentration in ICON-ART results in too short lifetimes of the tracers and that is why the lifetime of the species is parametrised in another way for stratospheric conditions.

However, the loss rate of acetone with Eq. (12) is also realistic above the UTLS region due to the photolytic reactions (R10) and (R11).

Therefore, another mechanism is applied above the UTLS region (indicated by the dashed blue line in Fig. 9) only if no other term is added to the loss rate. The lifetime of $\text{CH}_4$ is parametrised pressure-dependent like in the Integrated Forecast System (IFS, Simmons et al., 1989). In this parametrisation, the $\text{CH}_4$ lifetime in the troposphere is effectively infinite and decreases

for pressure below $100\,\text{hPa}$, e.g. it is $2000\,\text{days}$ at a pressure of $10\,\text{hPa}$. The CO lifetime is parametrised in the same way as in the KASIMA model (Karlsruhe SImulation model of the Middle Atmosphere) which also depends on pressure, only (Ruhnke et al., 1999; Kouker et al., 1999). The CO lifetime in this parametrisation in an altitude of $100\,\text{hPa}$ is about $1\,\text{year}$ and in $10\,\text{hPa}$ it is $25\,\text{days}$. The formulae of these two lifetime parametrisations have been published by Stassen (2015). The lifetime of propane is set globally to $14\,\text{days}$ (Rosado-Reyes and Francisco, 2007).

In order to be able to investigate processes within the UTLS region, a threshold in $\text{CH}_4$ of $1\,\text{ppmv}\,(= 10^{-6}\,\text{mol mol}^{-1})$ is applied. This value ensures the OH mechanism to be used in the lowermost stratosphere.

In Fig. 9, the zonal maximum of the air pressure where the $\text{CH}_4$ VMR decreases below $1\,\text{ppmv}$ (blue dashed) is illustrated along with the zonal minimum of the WMO tropopause pressure (black solid). Additionally, the zonally averaged VMR of $\text{CH}_4$ at the tropopause is shown (red dotted) which ranges from $1.65$ (Southern Hemisphere) to $1.7\,\text{ppmv}$ (Northern Hemi-

sphere). Due to its relatively long tropospheric lifetime, $\text{CH}_4$ is well-mixed in the troposphere and the $\text{CH}_4$ VMR does not

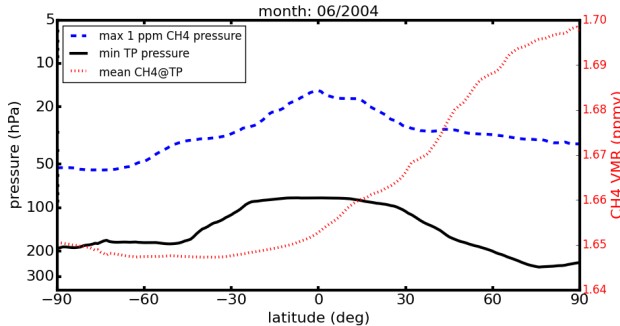

**Figure 9.** Zonal minimum of tropopause pressure, zonal maximum of $1\,\mathrm{ppmv}$ $CH_4$ pressure and zonal mean of $CH_4$ VMR at tropopause (right y-axis) for June 2004 of the OH-chem simulation (see Sect. 6). The $1\,\mathrm{ppmv}$ $CH_4$ pressure in each column is calculated as the air pressure of the model layer where $CH_4$ VMR decreases below $1\,\mathrm{ppmv}$.

decrease below $1\,\mathrm{ppmv}$. Above the tropopause, the $CH_4$ VMR decreases with height because of higher photolysis rates in the stratosphere.

As can be seen in Fig. 9, the lowest height where the $CH_4$ VMR decreases below $1\,\mathrm{ppmv}$ is clearly above the tropopause so that the OH mechanism is also applied in the lowermost stratosphere.

## 5  Measurements of acetone

We evaluate our simulations of acetone with observations from a) the KCMP tall tower measurements in Midwestern U.S. for seasonal and interannual variations, and b) the IAGOS-CARIBIC airborne measurements in the UTLS region in a similar way as recently published by Jöckel et al. (2016).

A suite of VOCs including acetone at the KCMP tall tower was measured by a proton transfer reaction mass spectrometer between July 2009 and August 2012 (Hu et al., 2013). The tower ($44.6886°$ N, $93.9728°$ W, $244\,\mathrm{m}$ height above ground) is located at rural area surrounded by croplands. Measurements were carried out $185\,\mathrm{m}$ above ground level, providing regional representativeness. The overall measurement uncertainty for acetone averages about $10\,\%$.

In the ongoing project Civil Aircraft for the Regular Investigation of the atmosphere Based on an Instrument Container (IAGOS-CARIBIC) a fully automated laboratory has been integrated into a modified cargo container (Brenninkmeijer et al., 2007). Measuring about 100 trace gases and aerosol parameters, the IAGOS-CARIBIC laboratory is regularly placed on-board a Lufthansa Airbus 340-600 passenger aircraft for up to six consecutive flights per month. The cruising altitude of the aircraft coincides with the UTLS region where measurements have been rare previously. Between 2005 and 2014, the flights took off in Frankfurt whereas the flights nowadays start in Munich in Germany to many intercontinental destinations. We use the acetone measurements from IAGOS-CARIBIC to compare them with the different innovations in ICON-ART (see Sect. 7.3). For our calculations, we use the data of the IAGOS-CARIBIC flights with the numbers 110 to 261 and 373 to 528. A statistic of the destinations of these flights can be found in Appendix B.

**Table 6.** Technical description of the simulations used in this study. For the used emission inventories see Table 3. Horizontal resolution for the simulations is R2B04 with an advective model time step of $460\,\mathrm{s}$. Output is given on model layers.

| simulation name | time range | output interval (in h) | short description |
|---|---|---|---|
| constL(megan-offl) | 2004-2015 | 23 | constant lifetime, offline emissions |
| constL(megan-onl,LAIsun) | 2004-2015 | 23 | constant lifetime, biogenic online emissions using $LAI_{\mathrm{sun}}$ |
| constL(megan-onl) | 2004-2015 | 23 | constant lifetime, biogenic online emissions |
| OH-chem(megan-offl) | 2004-2015 | 23 | tracer depletion with OH, offline emissions |
| OH-chem(megan-onl,LAIsun) | 2004-2015 | 23 | tracer depletion with OH, biogenic online emissions using $LAI_{\mathrm{sun}}$ |
| OH-chem(megan-onl) | 2004-2015 | 23 | tracer depletion with OH, biogenic online emissions |

## 6 Description of the ICON-ART simulations

We selected six simulations which are called constL(megan-offl), constL(megan-onl,LAIsun), constL(megan-onl), OH-chem(megan-offl), OH-chem(megan-onl,LAIsun) and OH-chem(megan-onl) hereafter. They are summarised in Table 6 from a technical point of view.

The simulations are performed with a horizontal resolution of R2B04 (characteristic length of about $160\,\mathrm{km}$). For output, they are interpolated to a regular $1°\mathrm{x}1°$ longitude-latitude grid. The lowest 46 of total 90 vertical layers are illustrated in Fig. 1. The advective model time step is set to $460\,\mathrm{s}$. All the simulations include an output interval of $23\,\mathrm{hours}$. With this interval, we are able to see the impact of OH on acetone at different times of day without using too many resources. Emissions as described in Table 3 are added to the tracers' VMR in the lowest model layer, see Sect. 3.1.3 for a discussion of this choice.

The meteorological variables such as temperature, pressure and three-dimensional wind as well as sea surface temperature and sea ice cover are initialised with ERA-Interim on 1 January 2004 at 00 UTC in order to cover the IAGOS-CARIBIC time range (2005 – 2015) with a spin-up period of one year for the chemical tracers. CO and $CH_4$ are initialised based on mean values provided by MACC reanalysis of January 2004 (Inness et al., 2013). $C_3H_8$ is initialised based on Pozzer et al. (2010). The initial volume mixing ratio of acetone is set globally to $1\,\mathrm{pptv}$. After initialisation ICON-ART runs freely.

*constL(megan-offl)*: The simulation using constant lifetime is the reference simulation for the other simulations. In this simulation, acetone lifetime is set globally to $28\,\mathrm{days}$. This is the mean value of the chemical lifetimes of Jacob et al. (2002), Arnold et al. (2005), Fischer et al. (2012) and Khan et al. (2015). The lifetime of $C_3H_8$ is set to $14\,\mathrm{days}$. That of CO and $CH_4$ are parametrised as described in Sect. 4.2 for the whole atmosphere.

    *constL(megan-onl,LAIsun)*: Simulation of online biogenic emissions of acetone is performed in this simulation where the
20 offline biogenic acetone emissions in constL(megan-offl) are replaced by MEGAN-Online $LAI_{\mathrm{sun}}$ (see Sect. 3.2).

    *constL(megan-onl)*: Simulation of online biogenic emissions of acetone is performed in this simulation where the offline biogenic acetone emissions in constL(megan-offl) are replaced by MEGAN-Online $LAI$.

    *OH-chem(megan-offl)*: In the simulation including the simplified OH chemistry, the mechanism as illustrated in Sect. 4 is used for depletion of the tracers and therefore replaces the constant lifetime of constL(megan-offl).

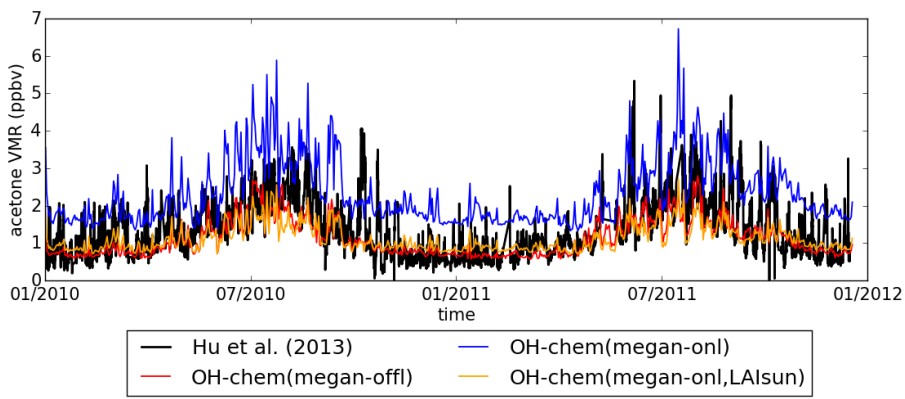

**Figure 10.** Measured (black) and simulated (red: OH-chem(megan-offl), blue: OH-chem(megan-onl), orange: OH-chem(megan-onl,LAIsun)) acetone VMR interpolated to the observation site in Minnesota (USA), see Hu et al. (2013). The measurement error of $10\,\%$ is not included in the figure.

*OH-chem(megan-onl,LAIsun)*: In this simulation, the offline biogenic acetone emissions in OH-chem(megan-offl) are replaced by MEGAN-Online $LAI_{\mathrm{sun}}$.

*OH-chem(megan-onl)*: Here, the biogenic emissions of acetone are replaced by MEGAN-Online $LAI$.

# 7   Results

## 7.1   Comparison of the ICON-ART simulations to ground-based measurements

Near-surface measurements of acetone are rare and no standard output of operational measurements. Data is available for several measurement campaigns such as by Schade and Goldstein (2006) or Fares et al. (2012) but only within one year. As our focus in this study is on the interannual variation of acetone in a climatological sense we here compare our simulations with the tall tower measurements in Minnesota, USA, performed by Hu et al. (2013).

Acetone was measured in a height of $185\,\mathrm{m}$ above ground so that the measurements should not be affected too much by local effects such as specific plants at the site. In addition, the measurements include a two years period from 2010 to 2011 and have already been used for a comparison to a global model (Hu et al., 2013).

In Figure 10, we show the results of the three OH-chem simulations together with the full observational time series. For the simulated time series, the horizontal grid point closest to the observation site is chosen and linearly vertically interpolated in geometric height to the measurement height. We cannot expect to simulate the full variability of the time series due to the coarse resolution of the simulation.

The acetone VMR of the OH-chem(megan-onl) simulation (blue line in Fig. 10) is by around factor 2 higher than the observed acetone VMR. Thus, Figure 10 suggests that the emissions of the MEGAN-Online $LAI$ scenario in Fig. 7 are unrealistically high.

In contrast to this, the simulated acetone VMR of OH-chem(megan-offl) and OH-chem(megan-onl,LAIsun) during the summer months (June to August) slightly underestimate the measurements. Altogether, these two time series resemble each other and are in good agreement with the observations. This confirms the previous results related to Figs. 7 and 8 where we have already discussed that the parametrisation of the $LAI$ according to Eq. (7) leads to results comparable with the inventory MEGAN-MACC.

Altogether, we conclude that the emission module performs well in comparison to the ground-based measurements. The average annual cycle is reflected in all our simulations. The overestimated OH-chem(megan-onl) simulation can be explained by a too large leaf area index (cf. Fig. 8). Especially the OH-chem(megan-offl) and OH-chem(megan-onl,LAIsun) simulations coincide well with the observed time series.

## 7.2 Profile of the acetone lifetime in the OH-chem simulations

In Figure 11, profiles of the annual mean acetone global lifetime according to Eq. (13) during the OH-chem(megan-offl) simulation are shown. For pressures higher than $900\,\mathrm{hPa}$, the photolysis rates in Eq. (12) get lower which means that the lifetime is dominated by the depletion with OH, only, leading to lifetimes up to $70\,\mathrm{days}$. In the troposphere and UTLS region, both mechanisms seem to have significant influence on the acetone lifetime. Due to the decrease in water vapour above the tropopause the production of OH by Reaction (R4) decreases. Additionally, the photolysis rates increase in the stratosphere for pressures below $50\,\mathrm{hPa}$ so that the influence of the OH depletion is negligible and the acetone lifetime decreases below one day.

When calculating the global annual mean tropospheric lifetime of acetone according to Eq. (13) in the OH-chem simulations, we derive a value (33 days) comparable to the one (35 days) by Arnold et al. (2005) who also used the definition of SPARC to calculate the acetone lifetime.

## 7.3 Comparison of the ICON-ART simulations with airborne measurements

Due to the seasonal variability in the biogenic emissions of acetone, its VMR in the mid-latitude UTLS region shows a seasonal cycle with maximum values above $1500\,\mathrm{pptv}$ during summer (Sprung and Zahn, 2010; Elias et al., 2011; Neumaier et al., 2014). This is shown in Fig. 12 where the acetone seasonal cycle $\pm3\,\mathrm{km}$ around the tropopause is derived from the IAGOS-CARIBIC measurements (panel a) and from the six ICON-ART simulations described in Sect. 6.

In the panels of Fig. 12, the simulated acetone VMR is linearly interpolated in pressure, longitude, latitude and time to the IAGOS-CARIBIC flights (see Eckstein et al., 2017). For calculation of the tropopause height we use the data sets which are most convenient for the measurements and simulations: the underlying temperature profiles for tropopause height in the IAGOS-CARIBIC measurements are derived from ERA-Interim profiles whereas the simulated tropopause height is calculated directly during runtime of ICON-ART (see Sect. 2.1). We limit the IAGOS-CARIBIC flights (and correspondingly the model data) to latitudes between 35 and $75°\,\mathrm{N}$ and exclude descents and ascents of the aeroplane by using data inside the pressure range of 280 and $180\,\mathrm{hPa}$ (similar to Jöckel et al., 2016).

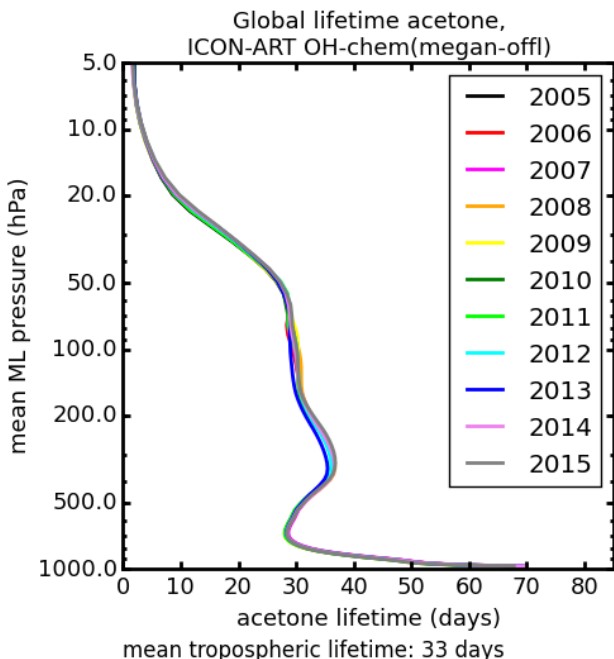

**Figure 11.** Global lifetime of acetone according to Eq. (13) in the OH-chem simulations averaged for each year. Definition of global lifetime by SPARC (2013) evaluated at each model layer.

Figure 12 demonstrates that the general annual cycle of acetone can be reproduced with ICON-ART. Maximum values in the acetone VMR of all ICON-ART simulations occur between June and August where also the measurements maximise. However, differences in the magnitude can be seen: For the simulations driven by offline emissions (left column) the maximum acetone VMR is underestimated by a factor of 3 with respect to the measurements.

As could be expected from Fig. 7, the annual cycles of acetone of constL(megan-onl,LAIsun) and OH-chem(megan-onl,LAIsun) are nearly identical with the respective offline emissions simulations except for slightly higher values in case of the LAIsun simulations. Thus, by parametrising the LAI according to Dai et al. (2004) the online biogenic emissions in ICON-ART are in good agreement with the offline data set MEGAN-MACC.

In contrast to this, MEGAN-MACC does not include a parametrised LAI. At least, there is no information given about it in

Sindelarova et al. (2014). This is why the resulting acetone VMR of the more than twice larger online emissions using $LAI$ have been suggested to be in agreement with MEGAN-MACC.

The acetone VMR around the tropopause using MEGAN-Online $LAI$ is shown in the rightmost column of Fig. 12. As these emissions are more than twice larger than the offline emissions the acetone VMR is increased in the UTLS region correspondingly. Thus, the differences with reference to observations are reduced but the highest values in the measurements can still not

be reached (around $1100\,\mathrm{pptv}$ compared to $1700\,\mathrm{pptv}$ in the measurements). Apart from the values in the maximum, Fig. 12d

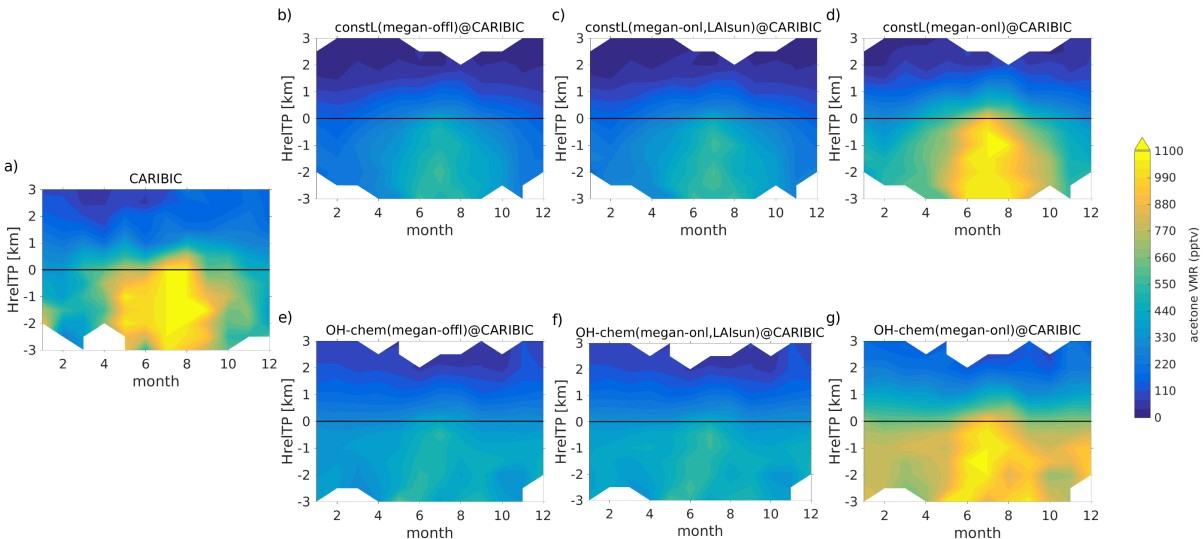

**Figure 12.** Annual cycles of the acetone VMR of (a) IAGOS-CARIBIC measurements and due to offline MEGAN-MACC (left column) and MEGAN-Online biogenic emissions with $LAI_{sun}$ (middle column) and $LAI$ (right column). Acetone VMR is shown $\pm 3\,\mathrm{km}$ around the WMO tropopause for constL (first row) and OH-chem simulations (second row). Data is limited to the mid-latitudes between 35 to $75°\,\mathrm{N}$ and to the pressure range between 180 and $280\,\mathrm{hPa}$. The acetone VMR in the IAGOS-CARIBIC measurements increases up to $1700\,\mathrm{pptv}$ in the maximum.

using MEGAN-Online $LAI$ combined with constant lifetime of acetone shows the best agreement with the observations in the upper troposphere: the acetone VMR during winter and "near-summer" only differs by $100\,\mathrm{pptv}$ or below.

To summarise the last paragraphs, we can reproduce the offline biogenic emissions data set MEGAN-MACC by parametrising the LAI in ICON with Eq. (7). This parametrisation ensures a more realistic treatment of the LAI with respect to the
5   biogenic emissions (Dai et al., 2004). However, the acetone VMR in the UTLS region is underestimated with respect to airborne measurements. The differences to the measurements are reduced if MEGAN-Online $LAI$ is used but the emissions are too high in this case (cf. Fig. 10). A discussion of this discrepancy follows in Sect. 7.4.

As already mentioned above, the global lifetime of acetone in the OH-chem simulations with a value of $33\,\mathrm{days}$ is in the same order of magnitude as in the constL simulations. That is why the maximum values in the acetone VMR in the OH-chem
10  simulations are comparable to the corresponding constL simulations. However, differences occur during winter where the clearly higher acetone lifetime of about $1.5\,\mathrm{years}$ in the OH-chem simulations increases the acetone VMR in the UTLS region. This value is a mid-latitudinal (35 to $75°\,\mathrm{N}$) average for the months December to February in 2005 to 2015 in contrast to the global annual average mentioned above.

The comparison of Fig. 12f with the observations demonstrates that the acetone VMR is overestimated by a factor of about 1.5 in the winter months December to February in the upper troposphere. In the lowermost stratosphere and especially above $2\,\mathrm{km}$ of the tropopause height, though, the acetone VMR is improved using the OH chemistry where the observations show higher VMRs than for the case of a constant lifetime (Fig. 12c).

## 7.4 Discussion

The findings in the UTLS region seem to be in conflict to the near-surface comparison of Fig. 10. In Fig. 12 the megan-onl simulations fit best to the measurements whereas the megan-offl and megan-onl,LAIsun simulations agree well with the ground-based observations. Several reasons could explain this discrepancy which will be discussed below:

1. uncertainty in the sources of acetone

2. uncertainty in the depletion of acetone

3. uncertainty in the model transport

4. uncertainty in the tropopause height

### 7.4.1 Uncertainty in the sources of acetone

As mentioned above acetone is directly emitted and is chemically produced by oxidation of monoterpenes, propane and isoalkanes (e.g. Jacob et al., 2002).

*Emission* data sets generally are highly uncertain. Sindelarova et al. (2014) estimated an uncertainty in MEGAN-MACC for isoprene emissions of globally $14\,\%$. For other VOCs, it could be even higher (e.g. $48.5\,\%$ by Williams et al., 2013). Oda et al. (2015) introduced a new method to calculate the emission uncertainty in a spatially dependent manner which resulted in a local uncertainty up to $200\,\%$ for $CO_2$. Based on Williams et al. (2013), we here assume the uncertainty in the acetone emissions to be in the order of $50\,\%$ which leads to a total uncertainty of $20\,\mathrm{Tg/yr}$.

*Acetone production due to oxidation of propane*: In the used emission inventories for propane (see Table 3) anthropogenic, biogenic and biomass burning emissions amount to $4.0$, $0.03$ and $1.7\,\mathrm{Tg(propane)/yr}$. The corresponding acetone production using the acetone yield of $0.736$ (see Eq. (14)) is about $5\,\mathrm{Tg(acetone)/yr}$. These values are close to that in the literature (Singh et al., 1994; Khan et al., 2015) although Singh et al. (1994) assume the global source of propane to be significantly higher in the order of $15$ to $20\,\mathrm{Tg(propane)/yr}$. A propane source in the order of $15\,\mathrm{Tg(propane)/yr}$ is also used in GEOS-Chem (Jacob et al., 2002; Fischer et al., 2012; Brewer et al., 2017). Due to the difference of about $10\,\mathrm{Tg(propane\ or\ acetone)/yr}$ in our configuration compared to that in GEOS-Chem we assume an uncertainty in this order for the acetone production in ICON-ART.

*Acetone production due to isoalkanes*: According to Jacob et al. (2002), it is in the order of $6\,\mathrm{Tg/yr}$.

*Acetone production due to monoterpenes*: We perform a test simulation using the monoterpenes in MEGAN-Online and calculate an acetone production of $8\,\mathrm{Tg/yr}$ using the same method as Brewer et al. (2017). They used a molar acetone yield

**Table 7.** Estimated uncertainty of acetone emissions and precursor species in $\mathrm{Tg(acetone)/yr}$.

| quantity | accounted for in ICON-ART2.0 | uncertainty (Tg/yr) |
|---|---|---|
| direct emissions | ✓ | 20 |
| oxidation of propane | ✓ | 10 |
| oxidation of isoalkanes | - | 6 |
| oxidation of monoterpenes | - | 8 |

by oxidation of monoterpenes of $0.116$. However, the explicit treatment of monoterpene chemistry could lead to a much higher acetone production, e.g. $46\,\mathrm{Tg/yr}$ by Khan et al. (2015).

Table 7 summarises our uncertainty estimates. The sum of all uncertainties is in the order of the acetone source itself which could explain the underestimation of acetone in the UTLS region.

The comparison to ground-based measurements at a site in Minnesota in Fig. **??** suggests that the emissions are well known in this specific region but the uncertainty could be higher in other parts of the world such as the tropical rain forests.

### 7.4.2 Uncertainty in depletion of acetone

Acetone is depleted by photolysis and reaction with OH, see Reactions (R9) to (R11). The used functions for reaction rates from Sander et al. (2011) include uncertainties. Due to their non-linear dependence on temperature small variations in the reaction rates can have great effect on the depletion of acetone.

In addition, for OH no VOC sink is included in our simplified approach. Especially isoprene is responsible for up to $70\,\%$ of the OH sink as measured by Hansen et al. (2014) and Rickly et al. (2017) close to the isoprene source in a forest.

Typical values of the OH reactivity during daytime measured by Hansen et al. (2014) in a boreal forest in northern Michigan (USA) are around $20\,\mathrm{s^{-1}}$. In our model, the OH reactivity over North America is around $5\,\mathrm{s^{-1}}$, so by factor of four lower than measured in this region.

This could lead to an overestimation of OH and therefore to a too fast depletion of acetone in its biogenic source regions, such as forests. Hence, this issue could also explain the underestimation of acetone in the UTLS region in Fig. 12 and future simulations should contain at least isoprene as main VOC sink of OH.

### 7.4.3 Uncertainty due to model transport

We performed a test simulation using another physics package in ICON. Especially, this physics package includes other treatment of convection and vertical diffusion. Differences to the simulations in Fig. 12 were clearly visible in the UTLS region including a shifted annual cycle similar as in the chemistry climate model EMAC (Jöckel et al., 2016). Further investigation of this issue is needed.

This example demonstrates that the model transport in atmospheric models also includes uncertainties which however are hard to quantify. With the resolution of R2B04 ($\sim 160\,\mathrm{km}$) used in this study, boundary layer convective processes could be underestimated leading to a too slow transport into the free troposphere.

### 7.4.4 Uncertainty in the tropopause height

For investigation of acetone in the UTLS region we use the WMO thermal tropopause height, derived from ERA-Interim or ICON-ART temperature. To investigate the uncertainty in the tropopause height another definition could be used for future simulations and especially for the measurements such as the definition derived from ozone as described by Sprung and Zahn (2010) or the dynamical tropopause height.

Higher or lower tropopause height in the measurements as well as in the model simulations could influence the climatologies in Fig. 12 since the acetone VMR depends on height, especially in the UTLS region.

### 7.4.5 Summary of the uncertainties

As could be shown in the previous sections the uncertainties in our simulations due to simplifications are quite large. These uncertainties can explain the mentioned conflict between Figs. 10 and 12.

To reduce the uncertainties further investigation is needed: The correct treatment and reduction of emission uncertainties in the inventories is current subject of research (e.g., Oda et al., 2015). The inclusion of monoterpenes and isoalkanes could improve the acetone source in our simulations. The inclusion of isoprene could lead to an improvement of OH in the source regions of acetone (Hansen et al., 2014). For evaluation of the model transport vertical profiles either from aircraft measurements or from satellites should be used in the future if available. In addition, the model transport near the surface could be investigated by introduction of a very short lived substance.

## 8 Conclusions and outlook

We introduce the recently developed module for including emissions from external data sources in ICON-ART. The module reads the data interpolated to the ICON grid, interpolates it to the simulation time and adds it to the trace gas volume mixing ratio in ICON-ART. For this, the number of lowest model layers of the emissions $n_{\mathrm{lev,emi}}$ has to be specified where we show a sensitivity test by varying this number. Differences only occur in the height of the emissions itself.

Therefore, the tracer mixing ratio above the emission height $n_{\mathrm{lev,emi}}$ is independent of the choice of $n_{\mathrm{lev,emi}}$. Since the aim of this study is the simulation of acetone in the upper troposphere and lowermost stratosphere (UTLS), we select $n_{\mathrm{lev,emi}} = 1$.

In addition, we demonstrate the online biogenic emission model MEGAN2.1 in the configuration as implemented in ICON-ART including two parametrisations of the leaf area index (LAI): the unparametrised LAI of ICON (MEGAN-Online $LAI$ scenario) and the LAI parametrised according to Dai et al. (2004) (MEGAN-Online $LAI_{\mathrm{sun}}$).

Emissions using in MEGAN-Online $LAI$ are twice larger than emissions of the offline emission inventory MEGAN-MACC. The emissions of MEGAN-Online $LAI_{\mathrm{sun}}$ are comparable to MEGAN-MACC in terms of global means and can therefore be used for investigating the influence of the diurnal cycle on acetone in the atmosphere.

Furthermore, we present a simplified parametrisation to deplete chemical species by reaction with OH. The OH concentra-
tion is calculated as steady state: it is produced by photolysis of ozone and reaction of the produced $O(^1D)$ with water vapour. It is depleted by reactions with $CH_4$ and CO.

With these new features, it is now possible to simulate volatile compounds (VOCs) with ICON-ART reliably. We illustrate this with acetone as one member of the VOCs.

Compared to ground-based measurements of Hu et al. (2013), our simulations generally show a comparable seasonal cycle.
Due to the higher emissions the acetone volume mixing ratio is overestimated with MEGAN-Online $LAI$. In contrast to this, the simulations with offline emissions and online biogenic emissions of MEGAN-Online $LAI_{\mathrm{sun}}$ are in good agreement with the observations with slight underestimation during summer. This demonstrates that the correct treatment of the LAI in MEGAN2.1 is crucial to get realistic results for online biogenic emissions. Further investigation of the representation of the emissions in MEGAN2.1 will follow in the future.

We also investigate the influence of the different features by comparing them to airborne measurements of the IAGOS-CARIBIC project in the UTLS region. With offline emissions and with online emissions of MEGAN-Online $LAI_{\mathrm{sun}}$ the acetone VMR in the UTLS region is underestimated by factor of 3. Correspondingly, it is increased by using the unparametrised LAI of ICON for online emissions. The simplified OH chemistry leads to a higher acetone lifetime especially during winter which results in an overestimation of the acetone VMR within December and February by a factor of about 1.5. On the other
hand, the acetone VMR in the lowermost stratosphere is improved by using the OH depletion mechanism.

Altogether, we show that the general acetone annual cycle is well represented in the model compared to the ground-based observations as well as to airborne IAGOS-CARIBIC measurements with a maximum during summer and a minimum during winter. Considering the acetone distribution in the lowest model layer we demonstrate that the presented emission module performs well. In addition, the calculated tropospheric acetone lifetime of $33\,\mathrm{days}$ is in good agreement with Arnold et al.
(2005) who used the same method to derive it. This value suggests that the new parametrisation of tracer depletion with OH is a good estimate of the OH concentration in the troposphere.

We also estimate the uncertainties in our simulations and further investigation should include the following aspects: adding further sources of acetone such as oxidation of higher VOCs, inclusion of isoprene as sink of OH, evaluation of the model transport by comparison to vertical profile measurements and using different definitions of the tropopause height.

**Code availability**

Currently the legal departments of Max Planck Institute for Meteorology (MPI-M) and DWD are finalising the ICON license. If you want to obtain ICON-ART you will first need to sign an institutional ICON license which you will get by sending a request to icon@dwd.de. In a second step you will get the ART license by contacting Bernhard Vogel (bernhard.vogel@kit.edu).

The method for including the emissions using the direct fluxes in the turbulence scheme, which was briefly discussed in this paper, will be added to a later release of ICON-ART.

## Appendix A: Predictor-corrector method

In this section, we explain the discretisation method for tracer concentration changes. We here refer to "concentration" as an abbreviation of number concentration (unit molecules per volume unit). Concentrations of tracers are determined by solving the following differential equation:

$$\frac{\partial c_i(x,t)}{\partial t} = P_i(x,t) - c_i(x,t)\, L_i(x,t) \tag{A1}$$

with $c_i$, $P_i$ and $L_i$ as concentration, chemical production and loss rate of tracer $i$. Concentration, chemical production and loss rate depend on location $x$ and time $t$. In ICON-ART version 1.0, this equation was discretised with the explicit Euler method (Rieger et al., 2015), omitting the index $i$ and the location dependence:

$$c_{t+\Delta t}^{(e)} = c_t^{(e)} + \left( P_t - c_t^{(e)} L_t \right) \Delta t \tag{A2}$$

Too low values of the tracer's lifetime can lead to solutions that do not converge to the differential equation (A1). Since fully implicit methods generally are expensive in computation resources, Seinfeld and Pandis (2012, pp. 1125–1126) suggest a two-step predictor-corrector discretisation method for solving Eq. (A1) which is discussed in this section. This method reasonably closes the gap between the low computation effort for explicit discretisation methods on the one hand and the accuracy and stability of implicit methods on the other hand.

Please note that the lifetime $\tau_t$ in this section is the reciprocal value of the loss rate: $\tau_t = 1/L_t$ (in contrast to the definition of SPARC used in the other sections).

Generally, Equation (A1) can be discretised implicitly as follows (Seinfeld and Pandis, 2012, pp. 1125–1126):

$$c_{t+\Delta t}^{(\text{ipc})} = \frac{c_t^{(\text{ipc})} \left( \tau_{t+\Delta t} + \tau_t - \Delta t \right) + 0.5\,\Delta t\,(P_{t+\Delta t} + P_t)\,(\tau_{t+\Delta t} + \tau_t)}{\tau_{t+\Delta t} + \tau_t + \Delta t} \tag{A3}$$

Lifetimes and productions of the next time step, denoted by index $t + \Delta t$, are not defined at time step $t$. That is why they have to be approximated before Eq. (A3) can be evaluated.

In a first step, called the predictor step, the new concentrations $c_*$ are approximated by assuming constant lifetime and production ($\tau_{t+\Delta t} = \tau_t$ and $P_{t+\Delta t} = P_t$):

$$c_* = \frac{c_t\,(2\,\tau_t - \Delta t) + 2\Delta t\,\tau_t\,P_t}{2\,\tau_t + \Delta t} \tag{A4}$$

In this study, these concentrations are calculated for $CH_4$, $CO$, propane and acetone. This is an inaccurate estimation of the concentrations of the next time step since lifetime and production both can vary within one time step (Seinfeld and Pandis, 2012, pp. 1125–1126). For improving accuracy, the lifetimes and productions of the next time step are approximated with the $c_*$ of Eq. (A4). For that purpose, $c_*$ is used for calculating a new OH number concentration, $[OH]_*$, as described in Sect. 4.1. In turn, with $[OH]_*$, the lifetimes and chemical productions of the next time step can be approximated, denoted as $\tau_*$ and $P_*$, respectively.

Then, the so-called corrector step can be executed in order to get the tracer concentrations of the next time step by replacing $\tau_{t+\Delta t}$ and $P_{t+\Delta t}$ in Eq. (A3) by their approximations $\tau_*$ and $P_*$, respectively:

$$c_{t+\Delta t}^{(pc)} = \frac{c_t^{(pc)} (\tau_* + \tau_t - \Delta t) + 0.5 \, \Delta t \, (P_* + P_t) (\tau_* + \tau_t)}{\tau_* + \tau_t + \Delta t} \tag{A5}$$

If $\Delta t$ becomes larger than the expression $\tau_* + \tau_t$, this method also can get instable.

To illustrate this, consider the following example by assuming the chemical production $P$ in Eq. (A5) to be zero, i.e.: $P_* = P_n = 0$ and additionally $\tau_* + \tau_n - \Delta t < 0$. In this case, the concentration of the next time step $c_{t+\Delta t}^{(pc)}$ becomes negative which obviously shows that the numerical solution does not converge the physical solution of the differential equation of Eq. (A1).

That is why we use the fully implicit Euler method assuming constant lifetime and chemical production if the lifetime gets lower than $\Delta t$:

$$c_{t+\Delta t}^{(i)} = P_t \, \tau_t + \left( c_t^{(i)} - P_t \, \tau_t \right) \exp \left( -\frac{\Delta t}{\tau_t} \right), \quad \tau_t < \Delta t \tag{A6}$$

**Appendix B: Statistic of the used CARIBIC flights**

Here, we show the destinations of all the CARIBIC flights used in the results. The statistic can be found in Table 8. For this, we counted the return flights as one flight and did not count stopovers of the aircraft as separate flights.

*Acknowledgements.* We thank Christoph A. Keller (NASA) for giving us details about the treatment of emissions within GEOS-Chem. Many thanks also to Sergey Gromov (Max-Planck-Institute for chemistry) for the inspiring discussions during EGU General Assembly 2017. We acknowledge ECCAD for archiving and distributing the data. This study was performed on the computational resource ForHLR II funded by the Ministry of Science, Research and the Arts Baden-Württemberg and DFG ("Deutsche Forschungsgemeinschaft").

**Table 8.** Frequency of occurrence of the destinations in the CARIBIC flights used for the climatologies in Sect. 7. The total number of different flights is 113.

| destination | number of flights |
| --- | --- |
| Manila (Philippines)[a] | 21 |
| Chennai (India) | 18 |
| Caracas (Venezuela) | 11 |
| Kuala Lumpur (Malaysia)[b] | 8 |
| Sao Paulo (Brazil) | 8 |
| Vancouver (Canada) | 8 |
| Santiago (Chile)[c] | 7 |
| Seoul (South Korea) | 7 |
| San Francisco (USA) | 7 |
| Los Angeles (USA) | 4 |
| Tokyo (Japan) | 3 |
| Toronto (Canada) | 2 |
| Denver (USA) | 2 |
| Houston (USA) | 1 |
| Bogota (Columbia) | 1 |
| Rio de Janeiro (Brazil) | 1 |
| Beijing (China) | 1 |
| Cape Town (South Africa) | 1 |
| Mexico City (Mexico) | 1 |
| Hong Kong (Hong Kong) | 1 |

stopover in [a] Guangzhou, [b] Bangkok and [c] Sao Paulo

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
