# Peer review of "An emission module for ICON-ART 2.0: Implementation and simulations of acetone"

_Geoscientific Model Development, 2016_

## Referee Comment (RC1) · Anonymous Referee #1 · 21 Nov 2016

**Review of M. Weimer et al.: A new module for trace gas emissions in ICON–ART 2.0: A sensitivity study focusing on acetone emissions and concentrations**

The authors wrote an article presenting a new module for emissions of gas phase tracers in the framework of the ICON–ART model version 2.0. They describe a first application of their module in a sensitivity study of acetone concentrations near the upper–troposphere lower–stratosphere (UTLS) region. After a short general introduction to ICON, a detailed description of the algorithmic implementation and the procedure to apply gridded emission data sets to ICON–ART follows. Alternatively, they implemented the MEGAN model for biogenic volatile organic compounds. They briefly

describe the basic properties of the MEGAN model. A first application of the new emission module is a sensitivity study of acetone in the UTLS region. All simulations are described and the emission data sets discussed. They use a simplified chemistry that is given in detail.

The subject of the paper is appropriate for Geosciences Model Development, however, I can not recommend it for publication without major revision. In fact, this paper merely describes how gas phase tracer emissions are applied inside ICON–ART, but it does not discuss the consequences of the various possibilities how emissions can be applied in such models nor do they reference other models using similar emission modules. Emissions in chemical transport models are widely used and nothing new. The reader is disappointed not to learn anything about these different methods although the title promises to present a sensitivity study on emissions. For a scientific evaluation of the emissions itself, the paper is too technical and "thin" with respect to the emissions themselves, but for a methodological paper, it's not detailed enough in terms of the method. The short application to acetone is interesting but the emission module itself needs to be analysed in more detail.

General comment

Emissions of gas phase tracers, in particular surface emissions, can be brought into the model by at least three methods: (*i*) as a flux condition at the surface to vertical diffusion. In that c ase, a net flux on the surface is calculated by adding the dry deposition flux. The vertical diffusion distributes the tracer in the boundary layer. (*ii*) as a source term in the chemical kinetic equations in some appropriate model layers. (*iii*) as a tendency in some appropriate model layers near the surface. The authors chose method (*iii*) but should discuss the other methods also which are all associated with a certain operator splitting. What concerns me most is the fact that the criteria are unclear according to which the number of lowermost layers are chosen into which the

emissions are brought as a tendency. The authors should discuss this choice, prepare an appropriate sensitivity study varying the number of levels and perform a simulation with emissions given as a flux condition to the vertical diffusion equation. In this latter simulation, the resulting tendencies in the lowermost model layers should be compared with the number of model layers used in method (*iii*). It would be particularly interesting to see the seasonal variation. Discuss other chemistry general circulation models and what methods they use.

Specific comments

**p.2, l.26:** The ICON model is not really "in development" anymore since the NWP physics is used for operational weather forcast.

**p.3, Tab.1:** The "official ICON grid number": Give a citation here.

**p.3, l.1:** It should be clarified that Leuenberger et al. generalized the SLEVE coordinates

**p.3, l.5 ff:** The usual definition of volume mixing ratios is moles tracer per moles dry air. In ICON, tracers are defined per moist air for the horizontal transport. How do you treat these different definitions? How did you check mass conservation?

**p.5, Fig.2:** "reset simulation year from boundary year back to current year": It's not so clear what you mean here. You would like to say that the last year for which emissions are given is repeated for all years later than this year. By the way, this may even be error prone repeating the last year without warning.

**p.6, l.10:** Emissions are interpolated by a nearest neighbour interpolation. A flux conserving interpolation would sound more natural since you like to have the same amount of tracer going into the atmosphere irrespective of the used horizontal model resolution. Discuss this.

**p.9, l.14:** "for each time step" may sound like model time step here, but you mean times for which emissions are provided. Please, rephrase.

**p.7, Fig. 3:** Lines are cut and not completely displayed.

**p.7, l.5:** Fig. 3 shows LATEX not TEX code.

**p.7, l.6:** "the number of emission": is it emission sectors, emission types? Be more specific.

**p.7, l.7:** "...account only the number..." an "if" is missing. "...is then used globally": Explain better that "the number" is then a globally applied emission mass flux.

**p.8, l.2:** "emission date" may be misinterpreted as the date when emissions are applied (actually, they are applied in every time step), but "emission date" refers to the date for which emission fluxes are provided. Please, rephrase.

**p.8, l.9:** Just a remark: Is it necessary to have such a fixed time limit of 11–years? Volcanic emissions could be very irregular, farther away than 11–years and should not be interpolated?

**p.8, l.13:** "After interpolation the emission is converted to VMR ($C_{\mathrm{emi},i}$)": Rephrase "emission" to "emission flux"; furthermore, you mean a VMR tendency here, the symbol $c$ is normally used for molarity, so $dX_{\mathrm{emi},i}/dt$ would be the correct symbol. Devide eq. (2) by time.

**p.8, eq.(2):** In mathematics, sums $\sum$ are not counting backwards, write $\sum_{l=k_{\mathrm{lev}}-k_{\mathrm{emi}}+1}^{k_{\mathrm{lev}}}$. If $k_{\mathrm{emi}} = 1$, I guess that all emissions should go into the lowest level $k_{\mathrm{lev}}$, consequently, the sum has to go from $k_{\mathrm{lev}}$ to $k_{\mathrm{lev}}$. The lower limit needs a $+1$ therefore. Explain the symbols.

**p.9, l.1:** Explain the choice of $k_{\mathrm{emi}}$ in more detail.

**p.9, l.5:** Websites are not a good reference in general. If necessary, add in a footnote the date when you accessed the sites the last time. Similarly: p. 9, l. 15, p. 11, l. 25, p. 16 the footnote, p. 22, l. 3/5

**p.11, l.17:** Avoid to start a sentence with a mathematical symbol, in particular when it repeats the symbol of the last sentence.

**p.11, l.28:** "derived" instead of "defined".

**p.12, l.6:** "sunlit leaves" instead of "sun leaves"

**p.13, fig.6:** Explain better how you calculate the means. First, the word "global" is irritating, since you are calculating means over "S–America"? In fact, you calculated means over a rectangle in longitude and latitude with sea points contributing to the surface but not to the emissions.

**p.15, l.7:** "...due to Reactions (R5) and (R6)" you probably like to say that (R5) is the rate–determining step.

**p.17, l.20:** ICON contains several time steps, which one do you mean?

**p.17, l.20:** Output interval of 23 hours, meaning output at 0h, 23h (of next day), 22h (of next day), 21h (of next day), and so on?

**p.18, l.1:** Temporal variability of OH at which time scale?

**p.19, l.15/20/Fig. 8:** OH–chem(off): Better choose OH–chem(megan–off) or similar acronyms to make clear that it is not the OH–chemistry being switched off. Fig. 8: Show the free troposphere in an additional panel such that the interannual variability becomes visible.

[Figure]

**p.19, l.12 ff:** Your reasons for the underestimation are pure hypotheses and more confusing than explaining your results. The reader is lost what refers to your simulations and what is speculation. Make it clear where your considerations are general and what concerns your situation.

**p.20, fig.9:** Remove unnecessary axis and titles of the plots. Create one color bar for all panels instead. Instead of a pressure range, give the geometric altitude of the tropopause.

**p.20, l.3ff:** Describe your result first, then interpret/explain

**p.20, l.13:** Acetone life time of 1.5 years versus 28 days? This is worth an explanation.

**p.21, l.5 ff:** The conclusions should contain information about the choice of $k_{emi}$ and the sensitivity studies versus a flux condition in vertical diffusion.

**p.22, l.1–5:** Code availability: give a contact person only since there seems to be no icon license available at the site given (address not found!). The ART license is referring to a person anyhow.

**Appendix A:** "Concentration" should be either "molarity" or "number concentration" since concentration can be anything, mole fraction, volume mixing ratios, molality, molonity and the like. It's confusing to have a species index and a time step index on $c$ in the same section. Put $c_{i,t}(x)$ where $x$ refers to the location, $i$ to the species and $t$ is time. Write the equations for an arbitrary integration step from $t$ to $t + \Delta t$. From (A2) on you can, if you like, omit the index $i$. Mark the various solutions with superscripts $^{(e)}$ for explicit Euler and $^{(pc)}$ for the predictor–corrector method.

**p.23, l.9:** Better: "If $\Delta t$ becomes larger than...". What do you mean with "unstable"?

**p.27, l.8:** Give the page in Pandis/Seinfeld.

Technical corrections

**p.1, l.12:** "dominated concentrations": missing space

**p.2, l.28:** delete one of the "to"

**p.13, l.11:** "The tracer loss rate" or "the loss rate of the tracers"

**p.19, l.8:** "Maximum" instead of "Maximal"

---

## Referee Comment (RC2) · Anonymous Referee #2 · 14 Dec 2016

In this study, the authors have newly introduced an emission module and a simplified OH chemistry module into ICON-ART aiming at simulations of VOCs. The VOC targeted in this study is acetone. The emission module has two options: the one is offline, in which external emission data are prepared in advance and are read during simulation, and the other one is online, in which emission values are calculated during simulation. In this study, the online emission module of MEGAN2.1 is implemented to simulate biogenic acetone emissions at a diurnal cycle scale. The OH chemistry module includes reactions related to CO, CH4, and acetone. In the model evaluation, the authors have compared simulated acetone VMR values with those observed by IAGOS-CARIBC in the UT/LS region and they have claimed that the developed model

performs reliably.

I understand that the previous study of Rieger et al. (2015) first developed ICON-ART specifically for aerosols and this study has extended that for trace gases. The ICON itself is a relatively new model and its application to the atmospheric chemistry is interesting. Therefore, this study is suitable for Geoscientific Model Development. However, I recommend major revision on this manuscript before publication.

Major comments:

As the title says, the emission module is claimed as the new topic, but readers cannot agree with this. Both the offline and online emission modules employ commonly-used techniques and are nothing new. Furthermore, descriptions of the off-line emission module are too technical and not suitable in the main text. I recommend to move most of the descriptions in Section 3.1 to a supplementary document as a sort of manual. Only descriptions of emission inventories used and Fig. 5 may be left in the main text.

Comparing only with IAGOS-CARIBIC is not sufficient and more evaluation analyses are required. The evaluation only with the UT/LS data might be misleading, if the model vertical transport, which is often very uncertain, is wrongly simulated. Surface station data may be available and they should be compared with the simulated values in addition. Furthermore, I cannot understand why the authors limited the IAGOS-CARIBIC data to the mid-latitude UT/LS region. I think tropical data and vertical profiles (if available) are also useful to evaluate the overall performance of the model. Furthermore, one more result with MEGAN-Online LAIsun, which is newly introduced in this study, is needed to be shown in the sensitivity test.

Minor comments:

Title: As stated above, "a new module for trace gas emissions" seems inappropriate.

P.1, L.12: Insert a space between "dominated" and "concentrations"

Introduction: What is the benefit of using ICON for atmospheric chemistry studies?

Please discuss about that. Also, other previous studies in which similar icosahedral models (other than ICON) are used for atmospheric chemistry should be cited; for example,

Suzuki, K., T. Nakajima, M. Satoh, H. Tomita, T. Takemura, T. Y. Nakajima, and G. L. Stephens (2008), Global cloud-system-resolving simulation of aerosol effect on warm clouds. Geophys. Res. Lett., 35, L19817, doi:10.1029/2008GL035449.

Elbern, H., J. Schwinger, and R. Botchorishvili (2010), Chemical state estimation for the middle atmosphere by four‐dimensional variational data assimilation: System configuration, J. Geophys. Res., 115, D06302, doi:10.1029/2009JD011953.

Niwa, Y., H. Tomita, M. Satoh, and R. Imasu (2011), A three-dimensional icosahedral grid advection scheme preserving monotonicity and consistency with continuity for atmospheric tracer transport. J. Meteor. Soc. Japan, 89, 3, 255–268.

Goto, D., et al. (2015), Application of a global nonhydrostatic model with a stretched-grid system to regional aerosol simulations around Japan, Geosci. Model Dev., 8, 235-259, doi:10.5194/gmd-8-235-2015.

P.2, L.28: "to to" => "to"

P.4, L.9: What is the overbar of rho?

P.8, L.14: I cannot understand the summation in Eq. (2).

P.9, L.1-2: These sentences are not clear to me.

P.9, L.9: The biomass burning emission seems duplicated. The MACCity inventory includes biomass burning, while another explicit biomass burning data of GFED is also added.

P.11, L.7: "leaf area index" => "leaf area index (LAI)" P.11, L.8: Delete "(LAI)" P.11, L.11: "leaf area index" => "LAI"

P.13, L.5: Why is the online emission so much higher than the offline one, although they are made by the same MEGAN?

P.14, L.4: What of Sander at al. (2011) is used?

Section 4.2: Is this reaction method for the stratosphere similar to those of other models?

P.16, L11: "(IFS)" Please cite a paper and list it in Reference, not describing the URL in the footnote.

P.16, L.20-P.17,L.1: "The air pressure corresponding...in the CH4 VMR." This reason is not enough for the validity of using 1ppmv CH4 as the threshold.

P.16, L.14: "110 to 261 and 373 to 528" Are they flight numbers? And where did the aircraft fly to? Please clarify.

P.17, L.20-21: "All the simulations . . . in the tracer concentrations" is not clear to me.

Appendix A: Description of tau is needed somewhere.

---

## Author Comment (AC1) · 14 Feb 2017

**Answer to comment of referee #1**

**A new module for trace gas emissions in ICON-ART 2.0: A sensitivity study focusing on acetone emissions and concentrations**

M. Weimer, J. Schröter, J. Eckstein, K. Deetz, M. Neumaier, G. Fischbeck, L. Hu, D. B. Millet,
D. Rieger, H. Vogel, B. Vogel, T. Reddmann, O. Kirner, R. Ruhnke, and P. Braesicke

Dear referee,

Thank you for your review of the paper. In the following, you can find our answers to your comments which are in red. When we talk about the "concentration" we mean the "number concentration" (in number of molecules per volume unit), just to clarify the used expressions. In addition, we have changed the variable name of the number of model layers of the emissions from $k_{\mathrm{emi}}$ to $n_{\mathrm{lev,emi}}$ and we refer to this number several times in our responses.

**1 General comments**

**Emissions of gas phase tracers, in particular surface emissions, can be brought into the model by at least three methods: (*i*) as a flux condition at the surface to vertical diffusion. In that case, a net flux on the surface is calculated by adding the dry deposition flux. The vertical diffusion distributes the tracer in the boundary layer. (*ii*) as a source term in the chemical kinetic equations in some appropriate model layers. (*iii*) as a tendency in some appropriate model layers near the surface. The authors chose method (*iii*) but should discuss the other methods also which are all associated with a certain operator splitting.**

Since we aim to follow the process splitting strategy of ICON (Rieger et al., 2015) we decided not to include emissions according to method (*ii*). Method (*i*) can only be used for one of the physics packages of ICON: Either the physics package for numerical weather prediction (NWP) or for climate projections (ECHAM physics). In case of method (*iii*) the algorithm for including the emissions follows the process splitting strategy as well as it is compatible with both the NWP and the ECHAM physics package.

Nevertheless, we performed a sensitivity test by including the emission mass fluxes in the NWP turbulence scheme. If the same variables are used as input, methods (*i*) and (*iii*) differ below $0.1\,\%$ and therefore are equivalent if the emissions are included into the lowest model layer, see Figure 1 herein.

We added the following sentence in Section 3: "Because of our aim to follow the process splitting concept of ICON (Rieger et al., 2015) and in order to be compatible with ICON for both numerical weather prediction and climate projections the emission mass flux densities are converted to volume mixing ratio and added to the tracer volume mixing ratios."

[Figure]

Figure 1: Spatially averaged profiles of the acetone VMR for different methods: adding emission fluxes to vertical turbulent diffusion (orange), method described in paper for $n_{\mathrm{lev,emi}} = 1$ (emission height of $20\,\mathrm{m}$, black dotted) and 2 to 12 (emission height of $65\,\mathrm{m}$ to $\sim 1500\,\mathrm{m}$ above ground, green thin lines).

**What concerns me most is the fact that the criteria are unclear according to which the number of lowermost layers are chosen into which the emissions are brought as a tendency.**

We have added a paragraph in Section 3.1.3 including Figure 1 in this answer. In this new paragraph, we describe that we select $n_{\mathrm{lev,emi}} = 1$ because the shown profiles are nearly identical above the height of around $750\,\mathrm{hPa}$. Since our aim in the paper is the simulation of acetone in the UTLS region our results should be robust against other choices of $n_{\mathrm{lev,emi}}$.

Our change in the manuscript:
"To investigate the differences in changes of $n_{\mathrm{lev,emi}}$ we perform sensitivity simulations of acetone by varying $n_{\mathrm{lev,emi}}$ between 1 and 12. These simulations are based on constL(megan-offl), see Sect. 6. In Figure 5, profiles of the acetone VMR are shown for the different choices of $n_{\mathrm{lev,emi}}$.

In the case of $n_{\mathrm{lev,emi}} = 1$, no emissions are included in the layers above. For larger values of $n_{\mathrm{lev,emi}}$ the VMR in the lowermost model layer decreases subsequently since the emissions are distributed into a larger column.

Above the specified emission height, all profiles converge each other and above around $750\,\mathrm{hPa}$ the influence of varying $n_{\mathrm{lev,emi}}$ is negligible. Because of our aim to simulate acetone in the UTLS region, the choice of $n_{\mathrm{lev,emi}}$ should make no difference. That is why we simply select $n_{\mathrm{lev,emi}} = 1$ for all used offline emissions."

**The authors should discuss this choice, prepare an appropriate sensitivity study varying the number of levels and perform a simulation with emissions given as a flux condition to the vertical diffusion equation.**

We have done that (see above and Figure 1 herein).

**In this latter simulation, the resulting tendencies in the lowermost model layers should be compared with the number of model layers used in method (*iii*). It would be particularly interesting to see the seasonal variation.**

As described above, we could show that both methods are equivalent. To avoid confusion, we focus on the method that has been chosen as the default implementation for ICON-ART.

**Discuss other chemistry general circulation models and what methods they use**

We have increased the introduction section of the paper with respect to this:
"Different approaches to include emissions in atmospheric modelling have been developed in the past and are used in current chemistry climate models: In the limited area chemistry model WRF-chem (Grell et al., 2005) emissions are treated as production terms in the chemical equations (McKeen et al., 1991). Emissions can be prescribed as a flux condition in the vertical diffusion, as e.g. in the Community Atmosphere Model (Lamarque et al., 2012; Neale et al., 2013) which is part of the Community Climate System Model (CCSM, Gent et al., 2011). This method is also used for emissions in the planetary boundary layer in the GEOS-Chem model (Bey et al., 2001) including the HEMCO module (Keller et al., 2014). Emissions in higher altitudes are brought into GEOS-Chem as a tendency in the respective height of the emissions (C. A. Keller, pers. comm., 2017). The MESSy interface (Jöckel et al., 2005) incorporated e.g. in the EMAC model (Jöckel et al., 2006) gives the possibility to choose the used method for including emissions into the model: Either emissions are prescribed as flux condition as described above or the increase of the tracer mixing ratio is calculated and added to the tracer (Kerkweg et al., 2006). The latter method is also used in the coupled limited area model COSMO-ART (COSMO: COnsortium for SMall-scale MOdelling, ART: Aerosols and Reactive Trace Gases, Vogel et al., 2009)."

**2   Specific comments**

**p.2, l.26: The ICON model is not really "in development" anymore since the NWP physics is used for operational weather forcast.**

We have rephrased this sentence:
"ICON is a non-hydrostatic atmospheric model developed with the aim of providing a global model for both weather and climate (Wan et al., 2013; Zängl et al., 2015). Since January 2016, it is operationally used for global numerical weather prediction at German Weather Service (DWD). In July 2016, ICON also replaced the limited area model COSMO-EU (Baldauf et al., 2011) by a nested area over Europe."

**p.3, Tab.1: The "official ICON grid number": Give a citation here.**

We have cited the web site where the global grids currently can be downloaded from: http://icon-downloads.zmaw.de/dwd_grids.xml
On this web site, the grids are called "official" grids.

**p.3, l.1: It should be clarified that Leuenberger et al. generalized the SLEVE coordinates**

We have done that.

**p.3, l.5 ff: The usual definition of volume mixing ratios is moles tracer per moles dry air. In ICON, tracers are defined per moist air for the horizontal transport. How do you treat these different definitions? How did you check mass conservation?**

We have reformulated Eqs. (2) and (3) so that it should be clear that the moles of air are calculated by the ideal gas law using pressure and temperature values of ICON. As these equations are then independent of the explicit molar mass of the air, the moles of the air are that of moist air.
With respect to the second question: The mass conservation of the tracers in ICON was discussed by Zängl et al. (2015). Since ICON-ART uses the ICON tracer structure to calculate the tracers this is also valid for ICON-ART.

The conversion to VMR in Sect. 3.1.3 is now formulated in the paper as follows (of course with other equation numbers):
"Generally, the VMR is defined as fraction of the number of moles of the tracer (in our case the number of moles of the emission $\Delta n_i$) and the number of moles of (moist) air $n_{\text{air}}$:

$$\Delta X_{\text{emi},i} = \frac{\Delta n_i}{n_{\text{air}}} \tag{1}$$

The moles of the emission are calculated as the emission mass flux density $E_i$ multiplied by the advective model time step $\Delta t$ and the base area $A$ of the grid box and divided by the molar mass of the species $M_i$:

$$\Delta n_i = \frac{E_i \, A \, \Delta t}{M_i} \tag{2}$$

The emission flux can be included into one or more lowest model levels to be specified in the LaTeX table, see Fig. 3. In the following, we will refer to this number as $n_{\text{lev,emi}}$. The total number of model layers is stated as $n_{\text{lev}}$. In ICON, the lowest model layer has the highest index so that the index of the lowest model layer is $l = n_{\text{lev}}$. For calculating the number of moles of the air we sum up the moles of air of the lowest $n_{\text{lev,emi}}$ model layers using the ideal gas law:

$$n_{\text{air}} = \sum_{l=n_{\text{lev}}-n_{\text{lev,emi}}+1}^{n_{\text{lev}}} n_{\text{air},l} = \sum_{l=n_{\text{lev}}-n_{\text{lev,emi}}+1}^{n_{\text{lev}}} \frac{p_l \, V_l}{R^* \, T_l} = \frac{A}{R^*} \sum_{l=n_{\text{lev}}-n_{\text{lev,emi}}+1}^{n_{\text{lev}}} \frac{p_l \, h_l}{T_l} \tag{3}$$

Accordingly, $p_l$, $T_l$, $h_l$ and $R^*$ stand for pressure, temperature and geometric height of the grid box and the universal gas constant, respectively.

With Eqs. (2) and (3) the VMR tendency of the emission $\mathrm{d}X_{\mathrm{emi},i}/\mathrm{d}t$, which is added to the tracer, is calculated according to:"

$$\frac{\mathrm{d}X_{\mathrm{emi},i}}{\mathrm{d}t} \approx \frac{\Delta n_i}{n_{\mathrm{air}}\,\Delta t} = \frac{E_i\,R^*}{M_i} \cdot \left(\sum_{l=n_{\mathrm{lev}}-n_{\mathrm{lev},\mathrm{emi}}+1}^{n_{\mathrm{lev}}} \frac{p_l\,h_l}{T_l}\right)^{-1} \tag{4}$$

**p.5, Fig.2: "reset simulation year from boundary year back to current year": It's not so clear what you mean here. You would like to say that the last year for which emissions are given is repeated for all years later than this year. By the way, this may even be error prone repeating the last year without warning.**

We have adapted the figure accordingly. In the ICON output a message is given for each file that is read so that the user is able to trace which files are used during runtime.

**p.6, l.10: Emissions are interpolated by a nearest neighbour interpolation. A flux conserving interpolation would sound more natural since you like to have the same amount of tracer going into the atmosphere irrespective of the used horizontal model resolution. Discuss this.**

We have investigated the total mass fluxes of the emissions for different resolutions. The global mass fluxes differ below $1\,\%$.

We have included the following sentence in the manuscript: "This method also conserves the total emission fluxes reasonably with a maximum deviation of 1 % in case of R2B04 and a less deviation for the other resolutions of Table 1 (not shown)."

**p.6, l.14: "for each time step" may sound like model time step here, but you mean times for which emissions are provided. Please, rephrase.**

We have done that: "Therefore the emission data have to be split into separate files according to their validity time."

**p.7, Fig. 3: Lines are cut and not completely displayed.**

That it is why it is called "extract" in the figure description. In addition, the important part, i.e. the tracer emission metadata, is shown completely.

**p.7, l.5: Fig. 3 shows LATEX not TEX code.**

We have replaced TeX by LaTeX.

**p.7, l.6: "the number of emission": is it emission sectors, emission types? Be more specific.**

We wanted to say "the number of emission types". We have included this word in the paper.

**p.7, l.7: "...account only the number..." an "if" is missing. "...is then used globally": Explain better that "the number" is then a globally applied emission mass flux.**

We have rephrased this sentence: "The standard value is taken into account only if the number of emission types is zero. Then it is used as the globally applied emission mass flux density. Otherwise [...]"

**p.8, l.2: "emission date" may be misinterpreted as the date when emissions are applied (actually, they are applied in every time step), but "emission date" refers to the date for which emission fluxes are provided. Please, rephrase.**

We have rephrased this sentence: "The first task of the module during runtime is to find the two dates closest to the simulation time where emission are available in the dataset."

**p.8, l.9: Just a remark: Is it necessary to have such a fixed time limit of 11 years? Volcanic emissions could be very irregular, farther away than 11-years and should not be interpolated?**

The module was created for the treatment of gas phase emissions from emission inventories. The used value of about 11 years is an arbitrarily chosen stop criterion that is far beyond the time resolution of commonly used inventories such as MACCity, MEGAN-MACC and GFED3. Volcanic emissions are treated in another way in ICON-ART (Rieger et al., 2015).

**p.8, l.13: "After interpolation the emission is converted to VMR ($C_{\mathrm{emi,i}}$): Rephrase "emission" to "emission flux"; furthermore, you mean a VMR tendency here, the symbol $c$ is normally used for molarity, so $dX_{\mathrm{emi,i}}/dt$ would be the correct symbol. Devide eq. (2) by time.**

We have included "flux" and reformulated Eq. (2) with respect to this (see above).

**p.8, eq.(2): In mathematics, sums are not counting backwards. [...]**

We have changed that.

**p.9, l.1: Explain the choice of $k_{\mathrm{emi}}$ in more detail.**

We have included a paragraph discussing this choice (see above in our answers to the General Comments).

**p.9, l.5: Websites are not a good reference in general. If necessary, add in a footnote the date when you accessed the sites the last time. Similarly: p. 9, l. 15, p. 11, l. 25, p. 16 the footnote, p. 22, l. 3/5**

We have done that and where possible we have given another citation.

**p.11, l.17: Avoid to start a sentence with a mathematical symbol, in particular when it repeats the symbol of the last sentence**

We have adapted this sentence.

**p.11, l.28: "derived" instead of "defined".**

We have changed that.

**p.12, l.6: "sunlit leaves" instead of "sun leaves"**

We are aware that the notions "sun leaves" and "sunlit leaves" have different meanings: The term "sunlit leaves" (and "shaded leaves") is used for expressing that vegetation is either directly lit by sun or shaded by other vegetation, which is also used by Dai et al. (2004). The discrimination between "sun leaves" and "shade leaves" is a botanical discrimination.
We actually used the same terms as Dai et al. (2004, "sunlit leaves") and Guenther et al. (2012, "sun leaves") in the paper.

In the manuscript we now consistently follow the naming given by Dai et al. (2004) and rephrased the sentence:
"For standard conditions, we use the average Photosynthetic Photon Flux Density (PPFDS) of the values given by Guenther et al. (2012): $\text{PPFDS} = 125\,\mu\text{mol}\,\text{m}^{-2}\,\text{s}^{-1}$."

**p.13, fig.6: Explain better how you calculate the means. First, the word "global" is irritating, since you are calculating means over "S-America"? In fact, you calculated means over a rectangle in longitude and latitude with sea points contributing to the surface but not to the emissions.**

We have excluded the global means from this figure and included a table with the global mass fluxes (now Table 5 in the paper).

**p.15, l.7: "...due to Reactions (R5) and (R6)" you probably like to say that (R5) is the rate-determining step.**

Yes, we do. We have included this in the paper:
"Reaction 5 results in a cascade of fast reactions and finally in a production of CO and is the largest source for atmospheric CO (Jacob, 1999; Boucher et al., 2001; Seinfeld and Pandis, 2012,

pp. 46–47). Since Reaction 5 is the reaction with lowest reaction rate of this cascade the chemical production of CO can be estimated as follows:"

**p.17, l.20: ICON contains several time steps, which one do you mean?**

In the documentation of ICON, it is called the "advective" time step which is equivalent to the time step of the fast physics (see Rieger et al., 2015). We have adapted this in the paper where we previously referred to the "model time step".

**p.17, l.20: Output interval of 23 hours, meaning output at 0h, 23h (of next day), 22h (of next day), 21h (of next day), and so on?**

Yes, that is right. Please see next comment for an explanation.

**p.18, l.1: Temporal variability of OH at which time scale?**

If the output interval is e.g. daily, we can only investigate OH concentrations at e.g. 00 UTC. However, the OH concentration strongly depends on the daily cycle and therefore also the compounds corresponding to the OH mechanism. That is why we chose an output interval less than daily.

We have adapted the sentence:
"All the simulations include an output interval of 23 hours. With this interval, we are able to see the impact of OH on acetone at different times of day without using too many resources."

**p.19, l.15/20/Fig. 8: OH-chem(off): Better choose OH-chem(megan-off) or similar acronyms to make clear that it is not the OH-chemistry being switched off. Fig. 8: Show the free troposphere in an additional panel such that the interannual variability becomes visible.**

We have adapted the simulation names to e.g. constL(megan-offl) and OH-chem(megan-onl). With respect to Figure 8: The interannual acetone lifetime in the free troposphere only differs by $1.7 \, \mathrm{days}$ in the maximum. That is why we think that the current figure is appropriate for the paper and no additional figure is needed.

**p.19, l.12 ff: Your reasons for the underestimation are pure hypotheses and more confusing than explaining your results. The reader is lost what refers to your simulations and what is speculation. Make it clear where your considerations are general and what concerns your situation.**

We have reconsidered our first argument because only the second and third ones of the listing are based on literature. Additionally, we have rephrased the remaining sentences:

"(1) We account for chemical production of acetone due to reaction of propane with OH but neglect the contribution of minor VOCs such as monoterpenes. The high impact of monoterpenes on acetone calculated by Khan et al. (2015) was recently challenged by Brewer et al. (2017). On the other hand, we neglect the weak uptake of acetone by the oceans and dry deposition which would decrease the acetone VMR slightly (Fischer et al., 2012; Khan et al., 2015). (2) [...]"

**p.20, fig.9: Remove unnecessary axis and titles of the plots. Create one color bar for all panels instead. Instead of a pressure range, give the geometric altitude of the tropopause.**

We have adapted the figure accordingly. As described in the text on p.19, l.5-7 the pressure range given in the figures refers to the pressure range where measurement data is considered.
We have added this information to the figure description.

**p.20, l.3ff: Describe your result first, then interpret/explain**

Due to a new sensitivity simulation, we have adapted the whole paragraph and included a description of the figure:
"The acetone VMR around the tropopause using MEGAN-Online $LAI$ is shown in the rightmost column of Fig. 12."

**p.20, l.13: Acetone life time of 1.5 years versus 28 days? This is worth an explanation.**

The main difference between these two values lies in their region and time scales they represent: The value of 28 days is a global annual mean value whereas the lifetime of 1.5 years stands for the northern winter mid-latitudinal average (35 to 75 °N).
In contrast to the global average, the low sun during (northern) winter decreases the photolysis rates related to OH and acetone in the mid-latitudes by one to two orders of magnitude. As the acetone loss rate in our simplified model is proportional to these photolysis rates, its lifetime increases accordingly.
We have included this sentence after p.20, l.13: "This value is a mid-latitudinal (35 to 75° N) average for the months December to February in 2005 to 2015 in contrast to the global annual average mentioned above."

**p.21, l.5 ff: The conclusions should contain information about the choice of $k_{emi}$ and the sensitivity studies versus a flux condition in vertical diffusion.**

We have added a paragraph in the conclusions describing our choice of $n_{lev,emi}$:

"[...] the number of lowest model levels of the emission $n_{lev,emi}$ has to be specified where we show a sensitivity test by varying this number. Differences only occur in the height of the emission itself.

Therefore, the tracer mixing ratio above the emission height $n_{\mathrm{lev,emi}}$ is independent of the choice of $n_{\mathrm{lev,emi}}$. Since our focus in the results is the comparison to measurements in the upper troposphere and lowermost stratosphere (UTLS), we choose $n_{\mathrm{lev,emi}} = 1$."

**p.22, l.1-5: Code availability: give a contact person only since there seems to be no icon license available at the site given (address not found!). The ART license is referring to a person anyhow.**

We have adapted the code availability paragraph by giving contact persons.

**Appendix A: "Concentration" should be either "molarity" or "number concentration" since concentration can be anything, mole fraction, volume mixing ratios, molality, molonity and the like. It's confusing to have a species index and a time step index on $c$ in the same section. Put $c_{i,t}(x)$ where $x$ refers to the location, $i$ to the species and $t$ is time. Write the equations for an arbitrary integration step from $t$ to $t + \Delta t$. From (A2) on you can, if you like, omit the index $i$. Mark the various solutions with superscripts $^{(e)}$ for explicit Euler and $^{(pc)}$ for the predictor-corrector method.**

We have clarified the expression concentration and adapted the indices of $c$: "We here refer to 'concentration' as an abbreviation of number concentration."

**p.23, l.9: Better: "If $\Delta t$ becomes larger than...". What do you mean with "unstable"?**

We now give an example in the paper where we assume the chemical production $P$ in Eq. (A6) to be zero, i.e.: $P_* = P_n = 0$ and additionally $\tau_* + \tau_n - \Delta t < 0$. In this case, the concentration of the next time step $c_{n+1}$ becomes negative which obviously shows that the numerical solution does not converge the physical solution of the differential equation.

**p.27, l.8: Give the page in Pandis/Seinfeld.**

We have added it for each citation in the main text.

**3    technical corrections**

We have corrected all the technical mistakes.

**References**

[revised manuscript text omitted]

---

## Author Comment (AC2) · 14 Feb 2017

**Answer to comment of referee #2**

**A new module for trace gas emissions in ICON-ART 2.0: A sensitivity study focusing on acetone emissions and concentrations**

M. Weimer, J. Schröter, J. Eckstein, K. Deetz, M. Neumaier, G. Fischbeck, L. Hu, D. B. Millet, D. Rieger, H. Vogel, B. Vogel, T. Reddmann, O. Kirner, R. Ruhnke, and P. Braesicke

Dear referee,

Thank you for your review of the paper. In the following, you can find our answers to your comments which are in red.

**1   General comments**

**As the title says, the emission module is claimed as the new topic, but readers cannot agree with this. Both the offline and online emission modules employ commonly-used techniques and are nothing new.**

We have adapted the title as follows: "An emissions module for ICON-ART 2.0: Implementation and simulations of acetone"

**Furthermore, descriptions of the off-line emission module are too technical and not suitable in the main text. I recommend to move most of the descriptions in Section 3.1 to a supplementary document as a sort of manual. Only descriptions of emission inventories used and Fig. 5 may be left in the main text.**

As described on the website of GMD, our goal is reproducibility: "[...] ideally, the description should be sufficiently detailed to in principle allow for the re-implementation of the model by others, so all technical details which could substantially affect the numerical output should be described"
In addition, Referee #1 requested an even more detailed description of the module. That is why we think that this section is appropriate for the main text.

**Comparing only with IAGOS-CARIBIC is not sufficient and more evaluation analyses are required. The evaluation only with the UT/LS data might be misleading, if the model vertical transport, which is often very uncertain, is wrongly simulated. Surface station data may be available and they should be compared with the simulated values in addition.**

We have compared the OH-chem simulations with the surface observations of Hu et al. (2013) and included it in the paper, now Sect. 7.1, and discussed the results.

**Furthermore, I cannot understand why the authors limited the IAGOS-CARIBIC data to the mid-latitude UT/LS region. I think tropical data and vertical profiles (if available) are also useful to evaluate the overall performance of the model.**

We, of course, agree with this comment in principle. However, there are several issues: Firstly the PTRMS needs some time to stabilise, i.e. the first hour of the measurements after take-off generally is not a reliable measurement. Furthermore the PTRMS is switched off at $\sim 700\,\mathrm{hPa}$ to prevent damage of the turbo molecular pumps during landing.

As a second point, our aim was to create a climatology with a methodology similar to that shown in Jöckel et al. (2016).

Additionally, for a meaningful climatology we need sufficient number of measurement points. As the CARIBIC container is always mounted in Germany (Frankfurt or Munich), then flying to an intercontinental airport and coming back to Germany again, the data coverage over the mid-latitudes is much higher than over the tropics.

**Furthermore, one more result with MEGAN-Online LAIsun, which is newly introduced in this study, is needed to be shown in the sensitivity test**

We have included this test and have discussed the results in Sect. 7.3:

"As could be expected from Fig. 7, the annual cycles of acetone of constL(megan-onl,LAIsun) and OH-chem(megan-onl,LAIsun) are nearly identical with the respective offline emissions simulation except for slightly higher values in case of the LAIsun simulations. Thus, by parametrising the LAI according to Dai et al. (2004) the online biogenic emissions in ICON-ART are in good agreement with the offline data set MEGAN-MACC."

**2   Minor comments**

**Title: As stated above, "a new module for trace gas emissions" seems inappropriate.**

We have adapted the title (see above).

**P.1, L.12: Insert a space between "dominated" and "concentrations"**

We have changed this.

**Introduction: What is the benefit of using ICON for atmospheric chemistry studies? Please discuss about that. Also, other previous studies in which similar icosahedral models (other than ICON) are used for atmospheric chemistry should be cited, for example, Suzuki et al. (2008), Elbern et al. (2010), Niwa et al. (2011), Goto et al. (2015)**

We have increased the introductory part with respect to this and included the sentence:
"Recent work also includes the development of chemistry-climate models on icosahedral grids (Suzuki et al., 2008; Elbern et al., 2010; Niwa et al., 2011; Goto et al., 2015; Rieger et al., 2015)."

**P.2, L.28: "to to" => "to"**

We have changed this.

**P.4, L.9: What is the overbar of rho?**

It means that the air density is Reynolds-averaged (see Rieger et al., 2015).
We have included it in the paper.

**P.8, L.14: I cannot understand the summation in Eq. (2).**

We have corrected the equation and explained all the symbols (of course the numbers of the equations herein differ from that used in the paper):

"Generally, the VMR is defined as fraction of the number of moles of the tracer (in our case the number of moles of the emission $\Delta n_i$) and the number of moles of (moist) air $n_{\mathrm{air}}$:

$$\Delta X_{\mathrm{emi},i} = \frac{\Delta n_i}{n_{\mathrm{air}}} \tag{1}$$

The moles of the emission are calculated as the emission mass flux density $E_i$ multiplied by the advective model time step $\Delta t$ and the base area $A$ of the grid box and divided by the molar mass of the species $M_i$:

$$\Delta n_i = \frac{E_i \, A \, \Delta t}{M_i} \tag{2}$$

The emission flux can be included into one or more lowest model levels to be specified in the LaTeX table, see Fig. 3. In the following, we will refer to this number as $n_{\mathrm{lev,emi}}$. The total number of model layers is stated as $n_{\mathrm{lev}}$. In ICON, the lowest model layer has the highest index so that the index of the lowest model layer is $l = n_{\mathrm{lev}}$. For calculating the number of moles of the air we sum up the moles of air of the lowest $n_{\mathrm{lev,emi}}$ model layers using the ideal gas law:

$$n_{\mathrm{air}} = \sum_{l=n_{\mathrm{lev}}-n_{\mathrm{lev,emi}}+1}^{n_{\mathrm{lev}}} n_{\mathrm{air},l} = \sum_{l=n_{\mathrm{lev}}-n_{\mathrm{lev,emi}}+1}^{n_{\mathrm{lev}}} \frac{p_l V_l}{R^* T_l} = \frac{A}{R^*} \sum_{l=n_{\mathrm{lev}}-n_{\mathrm{lev,emi}}+1}^{n_{\mathrm{lev}}} \frac{p_l h_l}{T_l} \tag{3}$$

Accordingly, $p_l$, $T_l$, $h_l$ and $R^*$ stand for pressure, temperature and geometric height of the grid box and the universal gas constant, respectively.

With Eqs. (2) and (3) the VMR tendency of the emission $\mathrm{d}X_{\mathrm{emi},i}/\mathrm{d}t$, which is added to the tracer, is calculated according to:"

$$\frac{\mathrm{d}X_{\mathrm{emi},i}}{\mathrm{d}t} \approx \frac{\Delta n_i}{n_{\mathrm{air}}\,\Delta t} = \frac{E_i\,R^*}{M_i} \cdot \left( \sum_{l=n_{\mathrm{lev}}-n_{\mathrm{lev,emi}}+1}^{n_{\mathrm{lev}}} \frac{p_l\,h_l}{T_l} \right)^{-1} \tag{4}$$

**P.9, L.1-2: These sentences are not clear to me.**

We have separated the calculation of the number of moles of the emission from Eq. (2) and reformulated these sentences (see above). As can be seen, the number of moles $\Delta n_i$ is independent of the emission height $n_{\mathrm{lev,emi}}$. We have reformulated the sentence:

"This method conserves mass of the emission since the calculated moles of the emission $\Delta n_i$ are independent of the choice of $n_{\mathrm{lev,emi}}$ and therefore do not change if $n_{\mathrm{lev,emi}}$ is increased."

**P.9, L.9: The biomass burning emission seems duplicated. The MACCity inventory includes biomass burning, while another explicit biomass burning data of GFED is also added.**

We actually only use the anthropogenic dataset and have removed the "and biomass burning" in the paper.

**P.11, L.7: "leaf area index" => "leaf area index (LAI)" P.11, L.8: Delete "(LAI)" P.11, L.11: "leaf area index" => "LAI"**

We have changed that.

**P.13, L.5: Why is the online emission so much higher than the offline one, although they are made by the same MEGAN?**

The advantage of using online emissions lies in the much higher temporal resolution of the input parameters, in case of MEGAN especially the temperature. Thus, emissions are calculated every model time step in contrast to the offline emissions which usually have a monthly temporal resolution. Therefore, it is clear that differences in the emission output occur.

In addition, our configuration is different from that used by Sindelarova et al. (2014) as described in Sect. 3.1.4 and 3.2. The input parameters and metadata come from another model and we adapted the MEGAN model which is described in Sect. 3.2. Hence, although MEGAN in ICON-ART and in Sindelarova et al. (2014) are based on the same source code of Guenther et al. (2012), its implementation is model-specific.

We have included a new figure where the sensitivity of the MEGAN-Online emissions on LAI is demonstrated and discussed (in Section 3.2 in the paper):

"In order to investigate the influence of the parametrisation of $LAI$ by Eq. (7) we show in Fig. 8 the distributions of $LAI$ and $LAI_{\mathrm{sun}}$, together with its influence on the acetone emission. As expected, large values in $LAI$ (top panel) occur over the Amazon region in South America as well as in Central Africa where also the acetone VMR in Fig. 6 maximises. In addition, the forest areas in the east of Canada, northern Europe and Siberia show large values of the $LAI$. In these regions, the $LAI$ is in the order of 3 to $6\,\mathrm{m}^2\,\mathrm{m}^{-2}$.

For the used solar zenith angle of $10.3°$, the parametrisation according to Eq. (7) smoothes and reduces the LAI to values around $1\,\mathrm{m}^2\,\mathrm{m}^{-2}$ (Fig. 8B). Only for the less vegetated regions such as desserts (Sahara or Atacama), the distribution of $LAI_{\mathrm{sun}}$ shows nearly no response to the parametrisation of Dai et al. (2004).

In the MEGAN model the emission mass flux density is proportional to $LAI$ (Guenther et al., 2012). That is why the resulting emissions in MEGAN-Online (Fig. 8C) depend linearly on the LAI for each shown plant type. The highest sensitivity on LAI can be seen for broadleaves in the tropics. Thus, the parametrisation of the LAI according to Dai et al. (2004) can lead to a reduction of the emission in the order of factor 2 to 3 in these regions.

To conclude, the correct treatment of LAI is crucial to get realistic results of the emissions in MEGAN. The parametrisation according to Dai et al. (2004) leads to emission flux densities in the same order of magnitude as in the offline data set MEGAN-MACC (see Fig. 7). Further investigation of this will be presented in Sect. 7."

**P.14, L.4: What of Sander et al. (2011) is used?**

We have adapted the sentence: "Cross sections and quantum yields are given in a tabulated form originating from Sander et al. (2011) and interpolated on given pressure and temperature values of Cloud-J."

**Section 4.2: Is this reaction method for the stratosphere similar to those of other models?**

The OH parametrisation as described in Section 4.1 is only valid for tropospheric conditions. In the paper, we are interested in UTLS acetone which is mainly driven by emissions at the surface. As shown in Fig. 7 (of the non-corrected manuscript) our definition of the UTLS region ranges high enough so that the stratospheric chemistry should not really disturb the simplified OH chemistry mechanism.

**P.16, L11: "(IFS)" Please cite a paper and list it in Reference, not describing the URL in the footnote.**

We have cited it.

**P.16, L.20-P.17,L.1: "The air pressure corresponding ... in the CH4 VMR." This reason is not enough for the validity of using 1ppmv CH4 as the threshold.**

We have rephrased the whole paragraph to clarify this:

"In Fig. 9, the zonal maximum of the air pressure where CH4 VMR decreases below $1\,\mathrm{ppmv}$ (blue dashed) is illustrated along with the zonal minimum of the WMO tropopause pressure (black solid). Additionally, the zonally averaged VMR of $CH_4$ at the tropopause is shown (red dotted) which ranges from $1.6$ (Sounthern Hemisphere) to $1.68\,\mathrm{ppmv}$ (Northern Hemisphere). Due to its relatively long tropospheric lifetime, $CH_4$ is well-mixed in the troposphere and the $CH_4$ VMR does not decrease below $1\,\mathrm{ppmv}$. Above the tropopause, the $CH_4$ VMR decreases with height because of higher photolysis rates in the stratosphere.

As can be seen in Fig. 9, the lowest height where the $CH_4$ VMR decreases below $1\,\mathrm{ppmv}$ is clearly above the tropopause so that the OH mechanism is also applied in the lowermost stratosphere."

**P.17, L.14: "110 to 261 and 373 to 528" Are they flight numbers? And where did the aircraft fly to? Please clarify.**

Yes, they are the CARIBIC flight numbers. We have included a statistic of the destinations of the flights used for the climatologies. It can be found in Appendix B in the paper.

**P.17, L.20-21: "All the simulations ... in the tracer concentrations" is not clear to me.**

If the output interval is e.g. daily, we can only investigate OH concentrations at e.g. 00 UTC. However, the OH concentration strongly depends on the daily cycle and therefore also the compounds corresponding to the OH mechanism. That is why we chose an output interval less than daily.

We have rephrased this sentence: "All the simulations include an output interval of 23 hours. With this interval, we are able to see the impact of OH on acetone at different times of day without using too many resources."

**Appendix A: Description of tau is needed somewhere.**

We have added it.

**References**

Dai, Y., Dickinson, R., and Wang, Y.: A Two-Big-Leaf Model for Canopy Temperature, Photosynthesis, and Stomatal Conductance, J. Clim., 17, 2281–2299, doi:10.1175/1520-0442(2004)017<2281:ATMFCT>2.0.CO;2, 2004.

Elbern, H., Schwinger, J., and Botchorishvili, R.: Chemical state estimation for the middle atmosphere by four-dimensional variational data assimilation: System configuration, J. Geophys. Res.: Atmospheres, 115, doi:10.1029/2009JD011953, d06302, 2010.

Goto, D., Dai, T., Satoh, M., Tomita, H., Uchida, J., Misawa, S., Inoue, T., Tsuruta, H., Ueda, K., Ng, C. F. S., Takami, A., Sugimoto, N., Shimizu, A., Ohara, T., and Nakajima, T.: Application of a global nonhydrostatic model with a stretched-grid system to regional aerosol simulations around Japan, Geosci. Model Dev., 8, 235–259, doi:10.5194/gmd-8-235-2015, 2015.

Guenther, A., Jiang, X., Heald, C., Sakulyanontvittaya, T., Duhl, T., Emmons, L., and Wang, X.: The Model of Emissions of Gases and Aerosols from Nature version 2.1 (MEGAN2.1): an extended and updated framework for modeling biogenic emissions, Geosci. Model Dev., 5, 1471–1492, doi:10.5194/gmd-5-1471-2012, 2012.

Hu, L., Millet, D. B., Kim, S. Y., Wells, K. C., Griffis, T. J., Fischer, E. V., Helmig, D., Hueber, J., and Curtis, A. J.: North American acetone sources determined from tall tower measurements and inverse modeling, Atmos. Chem. Phys., 13, 3379–3392, doi:10.5194/acp-13-3379-2013, 2013.

Jöckel, P., Tost, H., Pozzer, A., Kunze, M., Kirner, O., Brenninkmeijer, C. A. M., Brinkop, S., Cai, D. S., Dyroff, C., Eckstein, J., Frank, F., Garny, H., Gottschaldt, K.-D., Graf, P., Grewe, V., Kerkweg, A., Kern, B., Matthes, S., Mertens, M., Meul, S., Neumaier, M., Nützel, M., Oberländer-Hayn, S., Ruhnke, R., Runde, T., Sander, R., Scharffe, D., and Zahn, A.: Earth System Chemistry integrated Modelling (ESCiMo) with the Modular Earth Submodel System (MESSy) version 2.51, Geosci. Model Dev., 9, 1153–1200, doi:10.5194/gmd-9-1153-2016, 2016.

Niwa, Y., Tomita, H., Satoh, M., and Imasu, R.: A Three-Dimensional Icosahedral Grid Advection Scheme Preserving Monotonicity and Consistency with Continuity for Atmospheric Tracer Transport, J. Meteorolog. Soc. Jpn. Ser. II, 89, 255–268, doi:10.2151/jmsj.2011-306, 2011.

Rieger, D., Bangert, M., Bischoff-Gauss, I., Förstner, J., Lundgren, K., Reinert, D., Schröter, J., Vogel, H., Zängl, G., Ruhnke, R., and Vogel, B.: ICON-ART 1.0 - a new online-coupled model system from the global to regional scale, Geosci. Model Dev., 8, 1659–1676, doi:10.5194/gmd-8-1659-2015, 2015.

Sander, S., Abbatt, J., Barker, J., Burkholder, J., Friedl, R., Golden, D., Huie, R., Kolb, C., Kurylo, M., Moortgat, K., Orkin, V., and Wine, P.: Chemical Kinetics and Photochemical Data for Use in Atmospheric Studies, Evaluation No. 17, JPL Publication 10-6, 2011.

Sindelarova, K., Granier, C., Bouarar, I., Guenther, A., Tilmes, S., Stavrakou, T., Müller, J.-F., Kuhn, U., Stefani, P., and Knorr, W.: Global data set of biogenic VOC emissions calculated by the MEGAN model over the last 30 years, Atmos. Chem. Phys., 14, 9317–9341, doi:10.5194/acp-14-9317-2014, 2014.

Suzuki, K., Nakajima, T., Satoh, M., Tomita, H., Takemura, T., Nakajima, T. Y., and Stephens, G. L.: Global cloud-system-resolving simulation of aerosol effect on warm clouds, Geophys. Res. Lett., 35, doi:10.1029/2008GL035449, 119817, 2008.

---

## Referee Report (RR1)

**Review of revision of M. Weimer et al.: A new module for trace gas emissions in ICON–ART 2.0: A sensitivity study focusing on acetone emissions and concentrations**

The authors submitted an improved version of their article about their module for trace gas emissions in ICON–ART 2.0. Although I am convinced that all improvements contribute a lot to make the article better understandable to the reader, their deeper analysis also showed some disadvantages of their method. The emission module was used to assess the acetone concentration near the tropopause region that is far away from the surface where emissions take place. The acetone lifetime is estimated to be between 15 and 60 days globally, meaning that acetone is rather well evacuated from the boundary layer into the troposphere. The authors demonstrated that the results do not depend on their choice of the operator splitting and the fact that they inject the emissions into the lowermost model layer. On the other hand, it is not clear whether this method will give good results for other, e.g. very short lived trace gas species. In that respect, I consider the article being very specific and I like to suggest that this is briefly discussed since the method is neither new nor really original.

Generally, I think that the article can be published when the following points are improved.

**General comment**

This article is meant to describe a general method of bringing trace gas emissions into the ICON model. As stated above, only a very specific application case (acetone) is discussed in detail. For this application, the method is suitable as shown by the authors. On the other hand, for very short lived trace species, this is not so clear. The authors should discuss this point and recommend tests of suitability. Since they implemented surface emissions as flux conditions for their sensitivity studies, it would be great if they would include this method as an alternative in the code that will be distributed.

For me, there is no reason why one would like to follow operator splitting as a principle. I would rather consider any operator splitting as a necessary evil than a fundamental principle.

**Specific comments**

**p.1,l.1:** I would prefer "An emission module" instead of "An emissions module".

**Fig.3:** Please, mention that lines may be cut, the reader is confused otherwise.

**p.9, l.15 ff:** As far as I understand ICON, the pressure is not everywhere a "pressure of moist air". At some places a quasi–hydrostatic pressure of dry air may be used. There is no problem as long as the same pressure is used in equations (4) and (5) of your revised article. The tracer mass conservation is assured for the advection scheme only. So, please, control all the relevant equations whether these are correct for both parts of ICON into which you implemented your emission module.

**Appendix A:** The Predictor–corrector method is exactly that of Pandis and Seinfeld. Put it into the supplementary online material since this is an article about emission modules.

---

## Author Response (AR2)

**Letter to the Editor**

**Old title: An emissions module for ICON-ART 2.0: Implementation and simulations of acetone**

**New title: An emission module for ICON-ART 2.0: Implementation and simulations of acetone**

**M. Weimer, J. Schröter, J. Eckstein, K. Deetz, M. Neumaier, G. Fischbeck, L. Hu, D. B. Millet, D. Rieger, H. Vogel, B. Vogel, T. Reddmann, O. Kirner, R. Ruhnke, and P. Braesicke**

Dear Dr Folberth,

Please find below our point-by-point responses to the referees' comments with respect to the revised manuscript along with a marked-up version of the changes in the manuscript. The relevant changes include:

1. Due to the comment of Referee #1 we removed the "s" of "emissions" in the title

2. We have included a section at the end of the results (Sect. 7.4) discussing the uncertainties in our simulations which could explain the discrepancy between Figs. 10 and 12 which was the major comment of Referee #2

3. We have added an outlook at the end of the conclusions

Kind regards and on behalf of all co-authors,
M. Weimer

**Answer to report with respect to the revised version by referee #1**

**An emission module for ICON-ART 2.0: Implementation and simulations of acetone**

M. Weimer, J. Schröter, J. Eckstein, K. Deetz, M. Neumaier, G. Fischbeck, L. Hu, D. B. Millet, D. Rieger, H. Vogel, B. Vogel, T. Reddmann, O. Kirner, R. Ruhnke, and P. Braesicke

Dear referee,

Thank you for your review of the revised manuscipt. Please find below our responses to your comments.

**1 General comment**

**This article is meant to describe a general method of bringing trace gas emissions into the ICON model. As stated above, only a very specific application case (acetone) is discussed in detail. For this application, the method is suitable as shown by the authors. On the other hand, for very short lived trace species, this is not so clear. The authors should discuss this point and recommend tests of suitability.**

We have compared the two methods with isoprene with a lifetime of 2.4 hours in our model. As can be seen in Figure 1, both methods are also equivalent for tracers with such a short lifetime.

We have added the following sentence after p. 5 l. 20: "[The statement of equivalence of the methods] also holds for very short lived substances such as isoprene (not shown)."

**Since they implemented surface emissions as flux conditions for their sensitivity studies, it would be great if they would include this method as an alternative in the code that will be distributed.**

For the new method, changes in the ICON code are necessary. However, the official release policy of ICON does not foresee the implementation of new features in past major releases. Only hotfixes are allowed according to the official release policy. Hence, we included the new method into the development version of the code and it will appear in future releases.
We have added a paragraph in the "code availability" section describing this: "The method for including the emissions using the direct fluxes in the turbulence scheme, which was briefly discussed in this paper, will be added to a later release of ICON-ART."

**For me, there is no reason why one would like to follow operator splitting as a principle. I would rather consider any operator splitting as a necessary evil than a fundamental principle.**

We, of course, agree with this comment in principle. We have adapted the sentence concerning this issue: "In order to follow the process splitting concept of ICON (Rieger et al., 2015) and

to be compatible with ICON for both numerical weather prediction and climate projections the emission mass flux densities are converted to volume mixing ratio and added to the tracer volume mixing ratios."

**2  Specific comments**

**p.1,l.1: I would prefer "An emission module" instead of "An emissions module".**

We have corrected that.

**Fig.3: Please, mention that lines may be cut, the reader is confused otherwise.**

We have added the following sentence to the figure description: "Please note that the header lines are cut. For details of the table content see text."

**p.9, l.15 ff: As far as I understand ICON, the pressure is not everywhere a "pressure of moist air". At some places a quasi-hydrostatic pressure of dry air may be used. There is no problem as long as the same pressure is used in equations (4) and (5) of your revised article. The tracer mass conservation is assured for the advection scheme only. So, please, control all the relevant equations whether these are correct for both parts of ICON into which you implemented your emission module.**

We have checked that and we use the same variables as for the definition of the tracers. The internally used pressure in ICON is the pressure of moist air (including all hydrometeors). Only for output, the pressure is hydrostatically integrated. Many thanks to Daniel Reinert (DWD) for giving us these details.

**Appendix A: The Predictor-corrector method is exactly that of Pandis and Seinfeld. Put it into the supplementary online material since this is an article about emission modules.**

The description in Seinfeld and Pandis is quite general with respect to how to calculate the "star" values (Eq. A4 and A5). In the paper, we specified the algorithm by Seinfeld and Pandis to the OH chemistry parametrisation. That is why we think that the text is appropriate for the appendix.

**References**

Rieger, D., Bangert, M., Bischoff-Gauss, I., Förstner, J., Lundgren, K., Reinert, D., Schröter, J., Vogel, H., Zängl, G., Ruhnke, R., and Vogel, B.: ICON-ART 1.0 - a new online-coupled model system from the global to regional scale, Geosci. Model Dev., 8, 1659–1676, doi:10.5194/gmd-8-1659-2015, 2015.

[Figure]

Figure 1: Profile of isoprene using conversion to VMR (orange) and surface flux condition method (black dotted) for including emissions.

**Answer to report with respect to the revised version by referee #2**

**An emission module for ICON-ART 2.0: Implementation and simulations of acetone**

M. Weimer, J. Schröter, J. Eckstein, K. Deetz, M. Neumaier, G. Fischbeck, L. Hu, D. B. Millet, D. Rieger, H. Vogel, B. Vogel, T. Reddmann, O. Kirner, R. Ruhnke, and P. Braesicke

Dear referee,

Thank you for your review of the revised manuscipt. Please find below our responses to your comments.

**1 General comment**

**I have found that the manuscript has improved from the previous version. Especially, the comparison with the ground-based measurements adds scientific value to this study. However, I encourage the authors to elaborate the simulation results and make careful discussion. In the comparison with the ground-based measurements, MEGAN-Online LAI overestimated the acetone VMR, while the MEGAN-Offline and the MEGAN-Online LAIsun do not. In contrast, MEGAON-Online LAI works better in the comparison with CARIBIC. Unfortunately, these results are inconsistent with each other. The authors attributed this to emission or chemical production. However, it does not provide readers with valuable information. The authors should discuss which is more likely. Furthermore, the model transport could be another cause and this should also be discussed.**

We now have estimated the uncertainties as follows:

- direct emissions (accounted for in our simulations): based on Williams et al. (2013), emissions have an uncertainty of $50\,\%$. As acetone emissions are in the order of $40\,\mathrm{Tg/yr}$ this results an uncertainty of $20\,\mathrm{Tg/yr}$.

- acetone production due to oxidation of propane (accounted for in our simulations): in the used emission inventories anthropogenic, biogenic and biomass burning emissions of propane account for $4.0$, $0.03$ and $1.7\,\mathrm{Tg(propane)/yr}$. The corresponding total acetone production is about $5\,\mathrm{Tg(acetone)/yr}$. These values are close to that in the literature (Singh et al., 1994; Khan et al., 2015) although Singh et al. (1994) assume the propane emissions to be significanty higher in the order of $15$ to $20\,\mathrm{Tg/yr}$ which is used in GEOS-chem (Jacob et al., 2002; Fischer et al., 2012; Brewer et al., 2017). Because of the difference of $10\,\mathrm{Tg/yr}$ between GEOS-chem and our configuration we assume an uncertainty in this order of magnitude for acetone production due to propane oxidation.

- acetone production due to oxidation of isoalkanes: According to Jacob et al. (2002) in the order of $6\,\mathrm{Tg/yr}$.

Table 1: Estimated uncertainty in the acetone source in $\mathrm{Tg/yr}$

| quantity | uncertainty (Tg/yr) |
|---|---|
| direct emissions | 20 |
| oxidation of propane | 10 |
| oxidation of isoalkanes | 6 |
| oxidation of monoterpenes | 8 |

- acetone production due to oxidation of monoterpenes: We have performed a first test using the monoterpenes of the MEGAN model and got an acetone production of $8\,\mathrm{Tg/yr}$ with the same method as Brewer et al. (2017) who calculated a source of around $6\,\mathrm{Tg/yr}$. In this method, a molar yield of acetone from monoterpenes of $0.116$ is used. The explicit treatment of monoterpene chemistry however could lead to a much higher acetone production, compare  $46\,\mathrm{Tg/yr}$ by Khan et al. (2015).

Table 1 summarises our uncertainty estimates. The sum of all uncertainties is in the order of the acetone source itself which could explain the underestimation of acetone in the UTLS region.

With respect to the model transport, we have performed a simulation with another physics package. This includes especially other treatment of convection and vertical diffusion. We could see clear differences and have therefore included another paragraph discussing the uncertainty of the model transport in the paper.

**2   Minor comments**

**Page 5, L27: "preformed" => "performed"**

We have corrected that.

**Page 7, L9: "validity time" => "valid time"?**

We think that "validity time" is correct.

**Page 25, L23-25: unclear to me**

As Sindelarova et al. (2014) do not use a parametrisation of the LAI in the data set MEGAN-MACC we expected that our MEGAN model without a parametrisation of the LAI should lead to similar results. However, our results of the parametrised $LAI_\mathrm{sun}$ compare well with MEGAN-MACC and the ground-based observations.

**Page 26, L26-27: "The emissions of MEGAN-Online LAIsun are comparable to MEGAN-MACC in terms of global means and can therefore be used for investigating the influence of the diurnal cycle on acetone in the atmosphere." I think the influence of the diurnal cycle has not been well investigated in this study.**

This is meant as outlook for the future. We have added "for future simulations" to the sentence.

20    reaction rate of this cascade the chemical production of CO can be estimated as follows:

$$P_{\text{CO}} = k_{\text{CH4}} \, [\text{OH}] \, [\text{CH}_4] \tag{11}$$

As an example, we will focus on acetone in the following. Acetone is depleted either by reaction with OH or by photolysis where two channels have to be considered:

$$\text{CH}_3\text{C}(\text{O})\text{CH}_3 + \text{OH} \quad \overset{k_{\text{acetone}}}{\longrightarrow} \quad \text{Products} \tag{R9}$$

25    $$\text{CH}_3\text{C}(\text{O})\text{CH}_3 + \text{h}\nu \quad \overset{J_{\text{acetone},1}}{\longrightarrow} \quad \text{CH}_3\text{CO} + \text{CH}_3 \tag{R10}$$

$$\text{CH}_3\text{C}(\text{O})\text{CH}_3 + \text{h}\nu \quad \overset{J_{\text{acetone},2}}{\longrightarrow} \quad 2\text{CH}_3 + \text{
[revised manuscript text omitted]